# TORC1-dependent translation drives chromatin remodeling during the germ-cell-to-maternal transition in Drosophila

Noor M Kotb[1,2,6], Gulay Ulukaya[3], Anupriya Ramamoorthy [2], Lina Seojin Park[2], Julia Tang[2], Dan Hasson[3,4,5] & Prashanth Rangan [2,5✉]

## Abstract

**Proper oogenesis requires a programmed transition from an undifferentiated germ-cell gene expression program to a maternal gene-expression state. While this process depends on the heterochromatin-mediated silencing of germ-cell genes, the upstream mechanisms that enforce this transcriptional shift remain unclear. Here, we uncover a translation-driven chromatin remodeling program that promotes oocyte fate in Drosophila. Through a loss of function screen, we identify TORC1 activity (Mio, Raptor), ribosome biogenesis (Zfrp8, Bystin, Aramis), and a translation factor (eEF1α1) as essential for silencing the germ-cell program. We show that TORC1 activity increases during oocyte specification, and that disruption of TORC1 activity, translation, or ribosome biogenesis during this window impairs heterochromatin maintenance at germ-cell gene loci. Polysome profiling reveals that Zfrp8 promotes translation of the nuclear pore component Nucleoporin 44A (Nup44A), whose function is independently required for chromatin organization and repression of a cohort of germ-cell genes. Taken together, our findings reveal that a transient increase in translation orchestrates chromatin remodeling to ensure commitment to oocyte fate.**

**Keywords** Translation; Ribosome; TORC1; Chromatin; Oocyte
**Subject Categories** Chromatin, Transcription & Genomics; Development; Translation & Protein Quality

## Introduction

The formation of gametes is essential for sexual reproduction. During oogenesis, germ cells or germline stem cells (GSCs) differentiate and undergo meiosis to produce mature oocytes (Spradling et al, 2011; Lehmann, 2012). Once specified, the oocyte accumulates maternal RNAs, which are crucial for supporting early embryonic development (Laver et al, 2015). The mechanisms underlying the transition from a GSC or germ cell-specific transcriptional program to a maternal transcriptional program remain poorly understood. Here, we use the term "germ cell program" to refer to the transcriptional state of undifferentiated germline cells prior to their differentiation into an oocyte, in contrast to the maternal program established during oocyte specification.

Drosophila oogenesis is marked by a well-characterized transition from GSCs to an oocyte. The GSCs reside in the germarium, where they undergo asymmetric division to produce one GSC and a cystoblast (CB) (Fig. 1A) (Spradling et al, 2001; Xie and Spradling, 1998). CBs differentiate by expressing *bag of marbles* (*bam*) and undergo four incomplete mitotic divisions, forming 2-, 4-, 8-, and eventually 16-cell cysts (McKearin and Ohlstein, 1995; Ohlstein and McKearin, 1997). GSCs and CBs are marked by spectrosomes that transition into elongated fusomes in developing cysts (Fig. 1A) (Lin et al, 1994; de Cuevas et al, 1996; de Cuevas and Spradling, 1998). At the 16-cell cyst stage, one cell becomes the oocyte, while the remaining 15 cells develop into nurse cells that support the growing oocyte and produce maternal components (Huynh and St Johnston, 2004). The 16-cell cyst is then encapsulated by somatic follicle cells to form an egg chamber that develops into a mature egg (Fig. 1A) (Nystul and Spradling, 2010).

The germ cell-to-maternal transition occurs during the cyst stages of oogenesis and is marked by the silencing of genes that are expressed in germ cells, GSCs and early cyst stages that we collectively term "germ cell genes" and by a coordinated upregulation of maternal transcripts (Fig. 1A) (Sarkar et al, 2023; DeLuca et al, 2020; Kotb et al, 2024; Blatt et al, 2021). Silencing of the germ cell genes *ribosomal small subunit protein 19b* (*rpS19b*) and *blanks* is frequently used as a molecular readout of the germ cell-to-maternal transition (McCarthy et al, 2022; Sarkar et al, 2023). Despite the precise timing of this transition, how the silencing of germ cell genes is coordinated with oocyte specification remains unclear.

[1]Department of Biomedical Sciences/Wadsworth Center, University at Albany State University of New York (SUNY), Albany, NY 12202, USA. [2]Department of Cell, Developmental, and Regenerative Biology, Black Family Stem Cell Institute, Icahn School of Medicine at Mount Sinai, New York, NY 10029, USA. [3]Bioinformatics for Next-Generation Sequencing (BiNGS) Core, Tisch Cancer Institute, Icahn School of Medicine at Mount Sinai, New York, NY 10029, USA. [4]Department of Oncological Sciences, Icahn School of Medicine at Mount Sinai, New York, NY 10029, USA. [5]Graduate School of Biomedical Sciences, Icahn School of Medicine at Mount Sinai, New York, NY 10029, USA. [6]Present address: Hologic Diagenode, 400 Morris Avenue, Suite 101, Denville, NJ 07834, USA. ✉E-mail: prashanth.rangan@mssm.edu

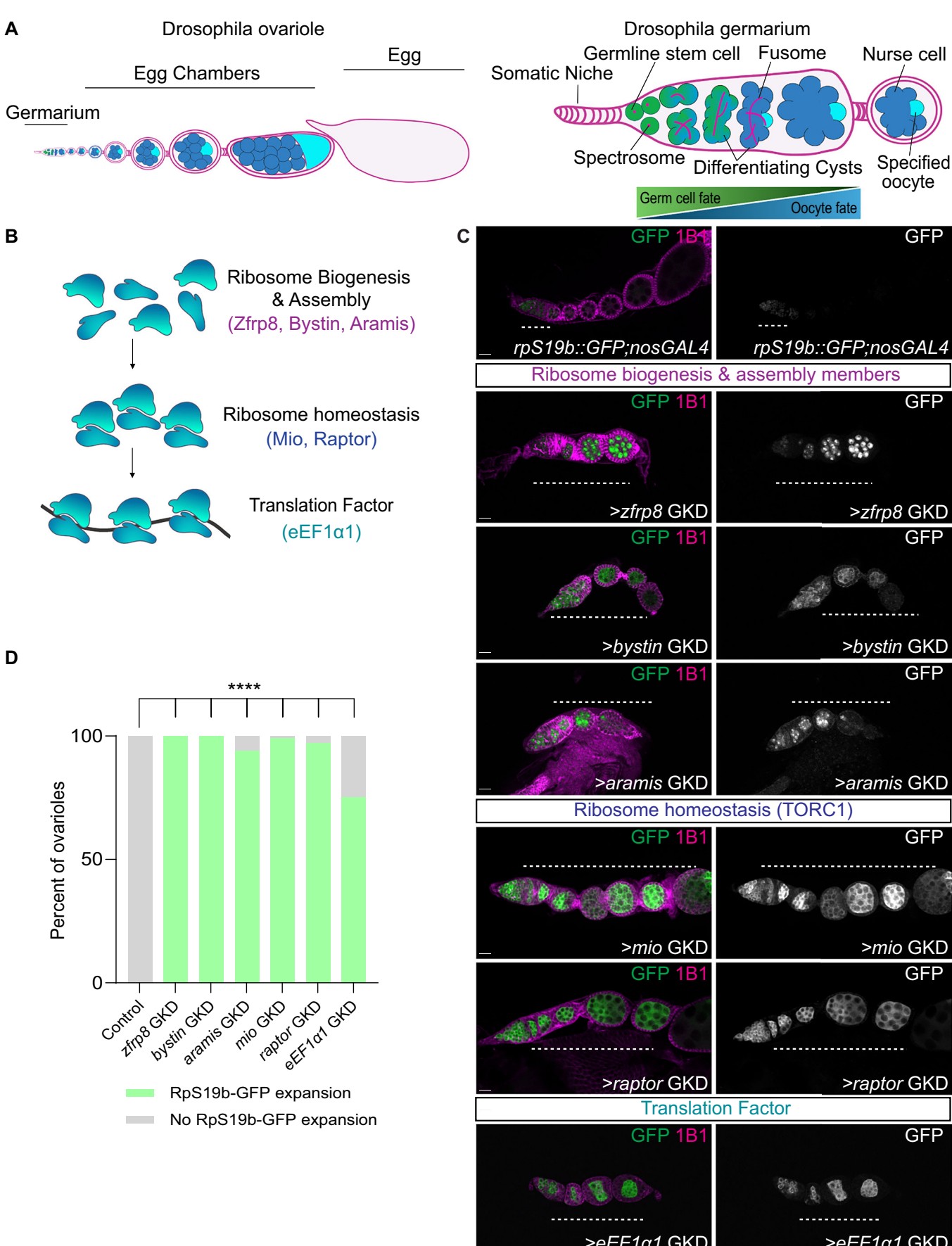

◀ **Figure 1.  Ribosome-level regulators are required for silencing the *rpS19b* reporter during oogenesis.**

(A) A schematic of a *Drosophila* ovariole consisting of a germarium containing early stages (green), egg chambers (blue) surrounded by somatic cells, and an egg. Egg chambers grow and produce an egg (light blue). The right panel shows a schematic depicting the *Drosophila* germarium. Near the somatic niche (light pink), are the germline stem cells (GSCs; green), dividing to give rise to daughter cells called cystoblasts (CBs). Both GSCs and CBs are identified by spectrosomes (red). CBs differentiate, giving rise to 2-, 4-, 8-, and 16-cell cysts (blue), marked by fusomes (red). Within the 16-cell cyst, a single cell initiates meiosis and becomes the oocyte (light blue), while the remaining 15 cells develop into nurse cells (dark blue). Genes associated with the germ cell program are active in the undifferentiated stages but decrease in expression as the oocyte is specified. Conversely, maternal genes become enriched in the later, differentiated stages. (B) Schematic of the genetic screen targeting regulators of ribosome biogenesis (Zfrp8, Bystin, and Aramis), ribosome homeostasis (Mio and Raptor), and translation (eEF1α1) to assess their role in gene silencing. (C) Ovariole of control *rpS19b::GFP;nosGAL4* stained for GFP (green) and 1B1 (magenta). In control, GFP is expressed in the undifferentiated stages and early cysts (white dashed line) and then silenced. Ovarioles from ribosome regulator germline knockdown (GKD) lines stained for GFP (green) and 1B1 (magenta). Loss of *zfrp8*, *bystin*, *aramis*, *mio*, *raptor*, and *eEF1α1* resulted in ectopic expression of RpS19b::GFP (white dashed line), failure of egg chamber growth, and oogenesis defects. Grayscale panels show GFP expression. (D) Quantification of ovarioles with expanded RpS19b::GFP expression. Statistics: two-tailed *t* test; $n = 50$ ovarioles per genotype. ns, $P > 0.05$; *$P < 0.05$; **$P < 0.01$; ***$P < 0.001$; ****$P < 0.0001$. Scale bars: 15 μm. Source data are available online for this figure.

The transcriptional silencing of germ cell genes requires the histone methyltransferase SET Domain Bifurcated Histone Lysine Methyltransferase 1 (SETDB1), which deposits Histone 3 Lysine 9 trimethylation (H3K9me3) to establish heterochromatin. (Clough et al, 2007; Rangan et al, 2011). Once silenced, these heterochromatic regions are demarcated by proteins such as Stonewall (Stwl) and anchored to the nuclear periphery via the nuclear pore complex (NPC) to maintain the silenced state (Sarkar et al, 2023; Kotb et al, 2024). Disruption of SETDB1, Stwl, or specific Nucleoporins (Nups) within the NPC results in aberrant expression of germ cell genes after the cyst stages, causing failed egg chamber growth and sterility (Maines et al, 2007; Clough et al, 2007; Sarkar et al, 2023; Kotb et al, 2024; Gigliotti et al, 1998). While these findings highlight the importance of chromatin reorganization during the germ cell-to-maternal transition, the upstream mechanisms driving this process remain unknown.

The Target of Rapamycin (TOR) pathway is a pivotal regulator of stem cell self-renewal and differentiation across many organ systems, including Drosophila oogenesis (Kim and Sabatini, 2004; Wilson et al, 2024; Sanchez et al, 2016; Martin et al, 2022). TOR activity integrates nutrient availability and energy status to promote anabolic processes such as ribosome biogenesis, protein synthesis, and cell growth (Laplante and Sabatini, 2012; Johnson et al, 2013). For instance, TOR activation relieves the translational repression of terminal oligopyrimidine (TOP)-containing mRNAs to support efficient ribosome production that promotes GSC differentiation in Drosophila (Martin et al, 2022). TOR exists in two distinct complexes: Target of Rapamycin Complex 1 (TORC1) and TORC2, each with specific functions and downstream targets (Zoncu et al, 2011). TORC1 activity is dynamically regulated by GAP Activity Toward Rags 1 (GATOR1) and GATOR2 during Drosophila oogenesis. GATOR1 promotes meiotic entry by inhibiting TORC1, whereas GATOR2 ensures TORC1 is reactivated in later stages to sustain differentiation (Wei et al, 2014). This balance between TORC1 inhibition and activation is essential for proper oogenesis. Loss of the GATOR2 components *missing oocyte* (*mio*) and *seh1* (also known as *Nup44A*) leads to arrested egg chamber growth and loss of oocyte fate (Iida and Lilly, 2004; Wei et al, 2014).

Intriguingly, a hypomorphic allele of the ribosomal protein gene *rpS2* (Cramton and Laski, 1994), loss of Zinc Finger RP 8 (Zfrp8), a protein that affects cytoplasmic stability of ribosomal proteins including RpS2 (Minakhina et al, 2016), and of GATOR2 components *mio* and *Nup44A* phenocopy the loss of *SETDB1*, *stwl*, and Nups, leading to egg chambers that do not grow and a loss

of oocyte fate (Iida and Lilly, 2004; Clough et al, 2007; Kotb et al, 2024; Maines et al, 2007; Gigliotti et al, 1998; Sarkar et al, 2023). Together, these findings raise the question of whether ribosome biogenesis and TORC1 activation orchestrate chromatin transitions across the cyst stage to ensure robust oocyte fate specification.

# Results

## Ribosome biogenesis, TORC1, and translation factors are required in the cyst stages for silencing *rpS19b* and *blanks*

To investigate the role of ribosome biogenesis in the germ cell-to-maternal transition, we depleted candidate genes that directly or indirectly regulate ribosome biogenesis and translation in the germline. These candidates included ribosome biogenesis regulators (*aramis*, *zfrp8*, *bystin*), TORC1 signaling components (*mio*, *raptor*), and translation elongation factor (*eEF1α1*) (Fig. 1B) (Minakhina et al, 2016; Martin et al, 2022; Fukuda et al, 2008; Iida and Lilly, 2004; Wei et al, 2014). We used the germline-specific *nanos* GAL4 (*nos*GAL4) driver to induce RNA interference (RNAi) in the *rpS19b::GFP* reporter line under moderate knockdown conditions, preventing excessively severe depletion to avoid GSC and cyst differentiation defects (Doren et al, 1998; McCarthy et al, 2022; Martin et al, 2022). Ovaries from control and ribosome regulator germline knockdown (GKD) lines were stained for GFP and 1B1, the latter of which marks somatic cell membranes, spectrosomes, and fusomes in the germline (Zaccai and Lipshitz, 1996). We found that GKD of ribosome biogenesis regulators *aramis*, *zfrp8*, and *bystin*, TORC pathway members, *mio* and *raptor*, and the translation elongation factor *eEF1α1*, resulted in egg chambers that failed to grow (Figs. 1C and EV1A). In control ovaries, RpS19b::GFP expression was confined to undifferentiated stages and repressed in the egg chambers. In contrast, ovaries with GKD of ribosome biogenesis factors, TORC1 regulators, and the elongation factor exhibited aberrant expression of RpS19b::GFP in egg chambers (Figs. 1C,D and EV1B). These findings suggest that regulators of ribosome biogenesis, TORC pathway members, and translation promote silencing of RpS19b::GFP in egg chambers and are required for egg chamber growth.

Germ-cell genes are initially silenced in the cyst stages, coinciding with elevated TORC activity (Fig. EV1C) (McCarthy et al, 2022; Sarkar et al, 2023; Martin et al, 2022; Sanchez et al, 2016). To determine whether reducing ribosome levels or

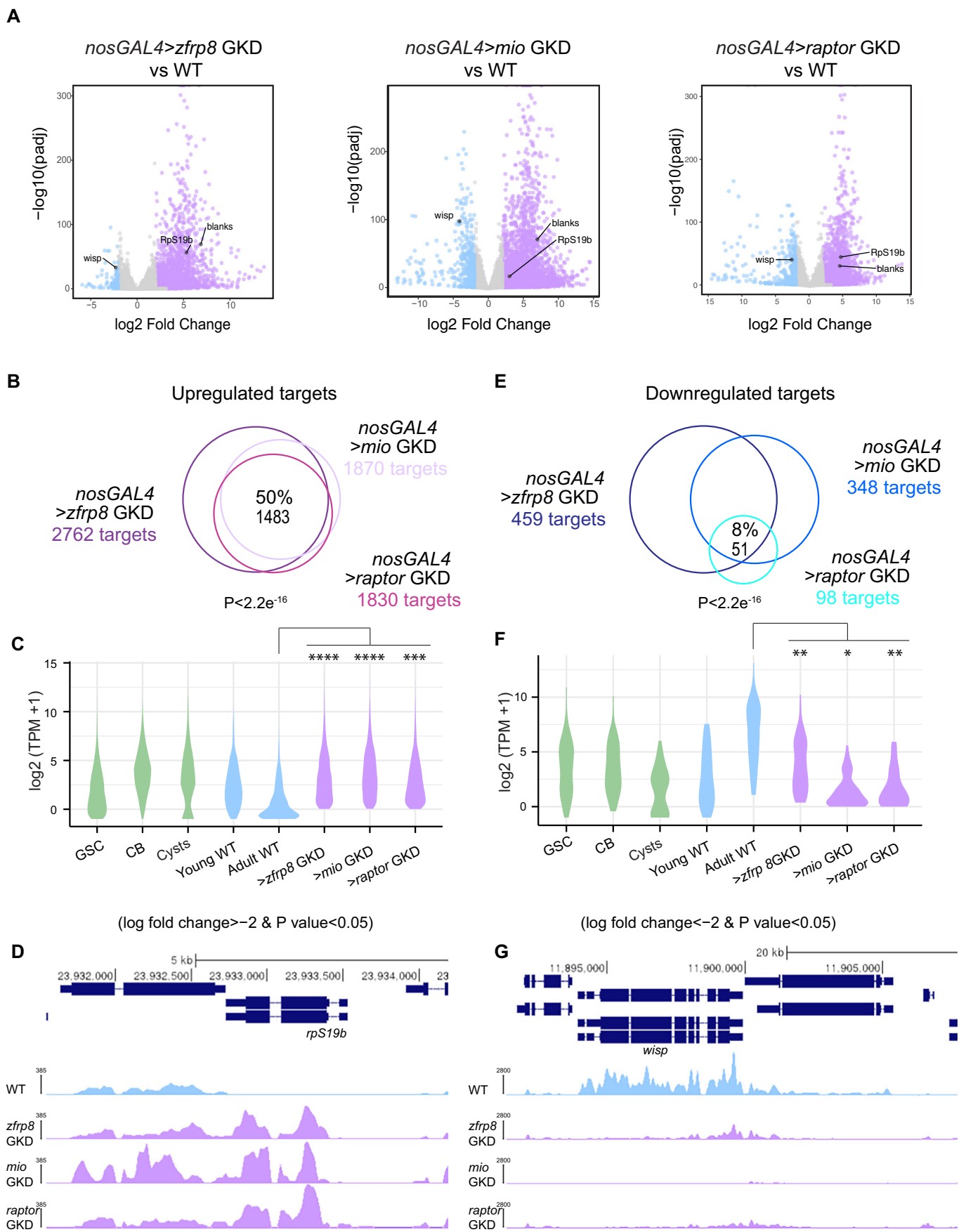

**A**

*nosGAL4>zfrp8* GKD vs WT

*nosGAL4>mio* GKD vs WT

*nosGAL4>raptor* GKD vs WT

**B** Upregulated targets

*nosGAL4 >mio* GKD 1870 targets

*nosGAL4 >zfrp8* GKD 2762 targets

50% 1483

*nosGAL4 >raptor* GKD 1830 targets

P<2.2e$^{-16}$

**E** Downregulated targets

*nosGAL4 >zfrp8* GKD 459 targets

*nosGAL4 >mio* GKD 348 targets

8% 51

*nosGAL4 >raptor* GKD 98 targets

P<2.2e$^{-16}$

**C**

(log fold change>−2 & P value<0.05)

**F**

(log fold change<−2 & P value<0.05)

**D**

*rpS19b*

WT

*zfrp8* GKD

*mio* GKD

*raptor* GKD

**G**

*wisp*

WT

*zfrp8* GKD

*mio* GKD

*raptor* GKD

**Figure 2. Ribosome-level regulators are required for silencing germ cell genes during oogenesis.**

(A) Volcano plots of $-\log_{10}$ adjusted p value vs. $\log_2$ fold change (FC) of nosGAL4 (control) vs >zfrp8 GKD, > mioGKD, and >raptor GKD ovaries showing significantly downregulated (blue) and upregulated (lilac) transcripts in zfrp8 GKD, mio GKD, and raptor GKD ovaries compared to control ovaries (FDR [false discovery rate] <0.05 and genes with twofold or higher change were considered significant). Differential expression was calculated using DESeq2 with two biological replicates per genotype (Benjamini–Hochberg correction). (B) Venn diagram of upregulated genes from RNA-seq of >zfrp8 GKD, >mio GKD, and >raptor GKD ovaries compared to controls. 50% of targets are shared upregulated targets, suggesting that ribosome-level regulators are required to upregulate a cohort of genes during oogenesis. Statistics: three-way overlap significance was computed by empirical randomization test (1,000,000 iterations); empirical $P < 10^{-6}$. (C) Violin plot of RNA levels of the shared upregulated targets between ribosome-level regulators in ovaries enriched for GSCs, CBs, cysts, and whole ovaries, showing that the upregulated targets are expressed up to the cyst stages and attenuated in whole ovaries. Statistics: a negative binomial regression model was used to estimate the average TPM count of "signature" genes between each "genotype". The TPM of each gene was used as the dependent variable, and "genotype" was the independent variable. Statistical comparisons between groups were performed using contrasts, and $P$ values were adjusted for multiple comparisons using the Benjamini–Hochberg procedure (FDR). The average TPM between groups was significantly different when $P$-FDR < 0.05. ns, $P > 0.05$; *$P < 0.05$; **$P < 0.01$; ***$P < 0.001$; ****$P < 0.0001$. Exact $P$ values ($P$-FDR) for upregulated targets in mutants compared to WT were: mio GKD vs WT ($P$-FDR $= 1.51 \times 10^{-5}$); raptor GKD vs WT ($P$-FDR $= 3.09 \times 10^{-4}$); and zfrp8 GKD vs WT ($P$-FDR $= 1.51 \times 10^{-5}$). (D) RPKM-normalized RNA-seq tracks showing that rpS19b is upregulated upon >zfrp8 GKD, >mio GKD, and >raptor GKD in dark purple compared to control (nosGAL4) (blue). (E) Venn diagram of downregulated genes from RNA-seq of nosGAL4 > zfrp8 GKD ovaries, mio GKD ovaries, and raptor GKD ovaries compared to controls. 8% of the targets are shared downregulated targets. Statistics: three-way overlap significance was computed by empirical randomization test (1,000,000 iterations); empirical $P < 10^{-6}$. (F) Violin plot of RNA levels of the shared downregulated targets between ribosome-level regulators in ovaries enriched for GSCs, CBs, cysts, and whole ovaries, showing that the downregulated targets are expressed at higher levels in the differentiated stages. Statistics: A negative binomial regression model was used to estimate the average TPM count of "signature" genes between each "genotype". TPM of each gene was used as the dependent variable, and "genotype" was the independent variable. Statistical comparisons between groups were performed using contrasts, and $P$ values were adjusted for multiple comparisons using the Benjamini–Hochberg procedure ($P$-FDR). The average TPM between groups was significantly different when $P$-FDR < 0.05. ns, $P > 0.05$; *$P < 0.05$; **$P < 0.01$; ***$P < 0.001$; ****$P < 0.0001$. Exact $P$ values ($P$-FDR) for downregulated targets compared to WT were: mio GKD vs WT ($P$-FDR $= 0.0159$); raptor GKD vs WT ($P$-FDR $= 0.00935$); and zfrp8 GKD vs WT ($P$-FDR $= 0.00206$). (G) RPKM-normalized RNA-seq tracks showing that wisp is downregulated upon >zfrp8 GKD, >mio GKD, and >raptor GKD in dark purple compared to control (nosGAL4) (blue). Source data are available online for this figure.

translation during this stage affects gene silencing, we used *bag of marbles* GAL4 (*bam*GAL4), which is active in the cyst stages, to drive RNAi (Chen and McKearin, 2003). We stained ovaries from control and cysts-specific GKD lines of *zfrp8*, *mio*, *raptor*, and *eEF1α1* for Blanks and 1B1 (Zaccai and Lipshitz, 1996). Compared to controls, the loss of *zfrp8*, *mio*, *raptor*, and *eEF1α1* in cyst stages resulted in egg chambers that failed to grow and exhibited persistent Blanks expression (Fig. EV1D,E). These findings suggest that ribosome biogenesis, TORC1, and translation are required during the cyst stages to silence Blanks and promote egg chamber growth.

## Components that support translation are required for silencing germ cell genes

To determine whether ribosome-level regulators broadly influence gene expression during germ cell-to-maternal transition, we first performed RNA-sequencing (RNA-seq) on control, *zfrp8*, *mio*, and *raptor* GKD ovaries, as these genotypes yielded sufficient material for analysis. We used a twofold cut-off (fold change (FC) = 2) and False discovery rate (FDR) < 0.05 to identify dysregulated genes (Fig. 2A). The overlapping set of genes in the three GKD ovaries reports on RNAs in the germline that are affected by the dysregulation of ribosome homeostasis. We detected 1483 genes that were upregulated in all three GKD ovaries (*zfrp8, mio*, and *raptor*) compared to controls, constituting 50% of all upregulated genes (Fig. 2B; Dataset EV1).

To determine when these 1483 genes are typically expressed during development, we examined RNA-seq libraries enriched for specific stages of oogenesis, including undifferentiated stages (GSCs and CBs), differentiating stages during oocyte specification (cysts), and differentiated stages (early and late egg chambers in young and adult flies, respectively) (McCarthy et al, 2022; Blatt et al, 2021). We found that the 1483 genes are typically expressed in undifferentiated stages and are repressed as differentiation progresses,

consistent with their classification as germ cell genes (Fig. 2C). Importantly, these genes, including *rpS19b* and *blanks*, were expressed in late egg chambers from GKD of *zfrp8*, *mio*, or *raptor* but not WT controls (Figs. 2C,D and EV2A). Thus, ribosome biogenesis and TORC1 activity is required to promote the silencing of germ cell genes at the onset of oocyte specification.

In addition, we detected 51 genes that were downregulated in the *zfrp8, mio*, and *raptor* GKD ovaries compared to controls, constituting 8% of all downregulated genes (Fig. 2E). Using the stage-specific RNA-seq libraries, we found that these 51 genes are highly expressed in egg chambers of WT ovaries but not GKD of *zfrp8*, *mio*, or *raptor* egg chambers (Fig. 2F). Notably, these dysregulated genes encode maternally deposited transcripts, such as *wispy* and *zelda* (Figs. 2G and EV2B). Therefore, ribosome biogenesis and TORC1 activity are also essential for upregulating a small subset of genes, including those that are required to launch the next generation.

## Ribosome biogenesis, TORC1 activity, and heterochromatin machinery converge on silencing of germ cell genes

The H3K9me3 histone methyltransferase, *SETDB1*, genome organization proteins such as *Nup154* (a component of the NPC), and the H3K27me3 histone methyltransferase *enhancer of zeste* (*e(z)*) are required for silencing of germ cell genes in differentiated egg chambers and for egg chamber growth (Sarkar et al, 2023; DeLuca et al, 2020). The phenotypic similarity between ribosome-level regulators and chromatin modulators suggested a potential link between ribosome biogenesis, TORC1, and chromatin regulation during oocyte development. To test this, we plotted $\log_2$FC of genes in *zfrp8*, *mio*, and *raptor* GKD with previously published datasets for *SETDB1*, *Nup154*, and *e(z)* GKD ovaries (Sarkar et al, 2023; DeLuca et al, 2020). Strikingly, 47% (658 genes) of the upregulated targets in ribosome regulators *zfrp8*, *mio*, and *raptor*

**A**

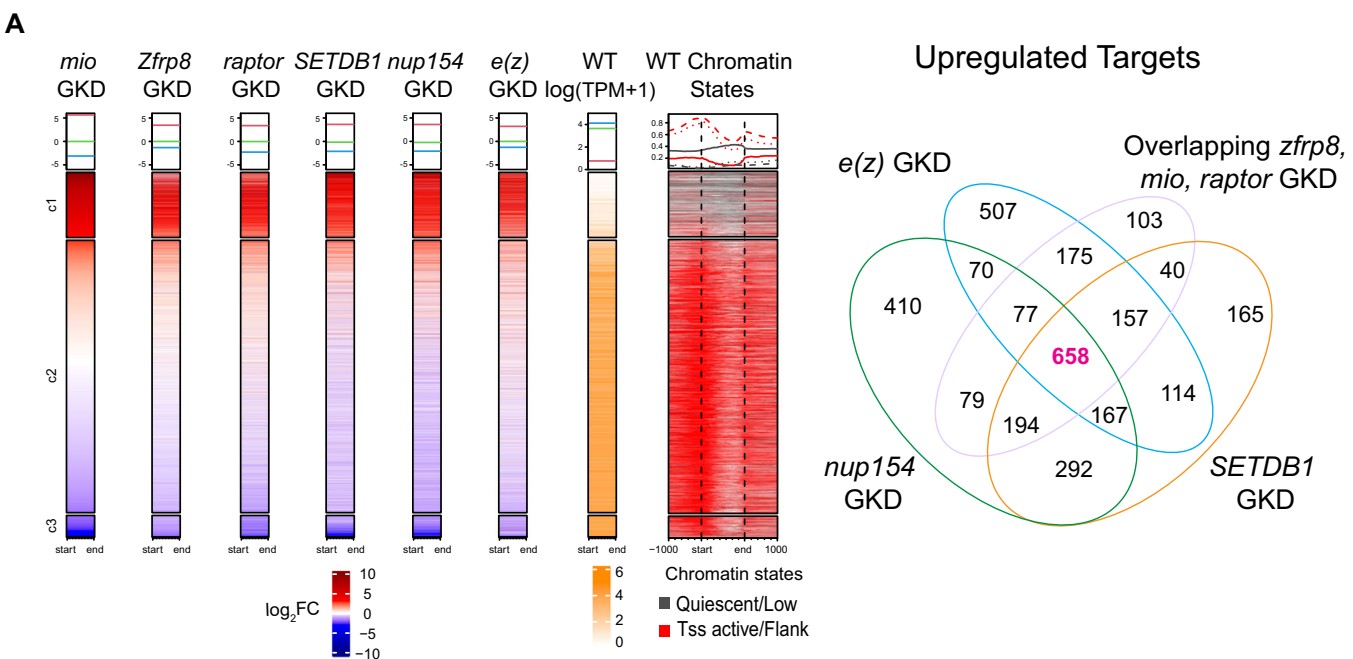

**B**

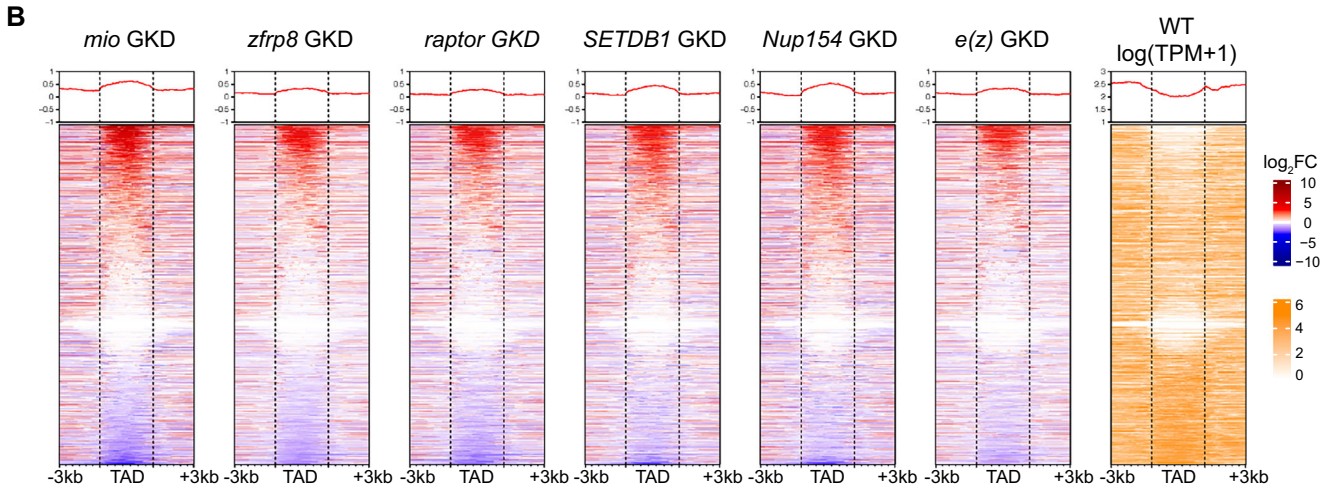

**C**          **D**

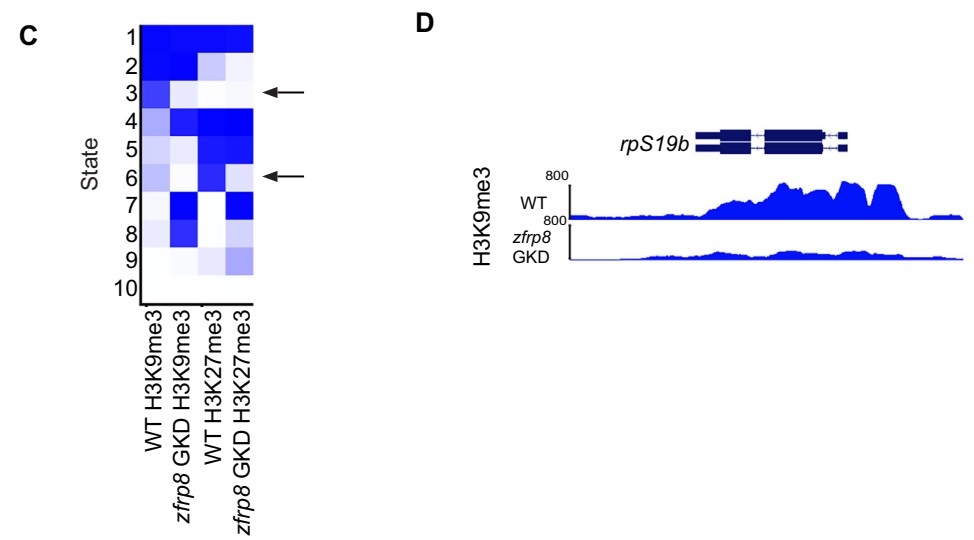

Figure 3. Ribosome-level regulators control a cohort of common targets with heterochromatin modulators.

(A) Heatmaps of log₂FC sorted genes *zfrp8* GKD, *mio* GKD, and *raptor* GKD along with *SETDB1* GKD, *Nup154* GKD, and *e(z)* GKD targets showing they all share a common cohort of upregulated and downregulated targets. Upregulated targets are shown in red and downregulated targets are shown in blue. Heatmaps show chromatin states for those targets in differentiated stages of wild-type ovaries. Venn diagram showing overlapping upregulated targets between *zfrp8* GKD, *mio* GKD, and *raptor* GKD with *SETDB1* GKD, *e(z)* GKD, and *Nup154* GKD. The overlap is 658 genes (47%) showing that ribosome-level regulators and heterochromatin regulators share a common cohort of upregulated targets during oogenesis. Statistics: Hypergeometric test *P* < 2.2e⁻¹⁶. (B) Heatmaps of log₂FC sorted for RNA-seq targets of ribosome regulators at TADs. Upregulated targets are shown in red, and downregulated targets are shown in blue. Heatmaps are shown for TADs as well as −3 kb and +3 kb around the start and end of TADs. Black dashed lines represent the start and end of TADs. The heatmaps are divided into three clusters: the upregulated target cluster, the nontarget gene cluster, and the downregulated target cluster. The heatmaps show that both upregulated genes and downregulated genes upon ribosome regulator GKD are present within TADs. (C) A chromatin model for H3K9me3 and H3K27me3 regulation in WT and >*zfrp8* GKD ovaries. In *zfrp8* GKD, H3K9me3 levels are reduced in state 3 while H3K27me3 is redistributed, indicating chromatin instability. These findings suggest that ribosome biogenesis is required for maintaining epigenetic silencing during oocyte specification. (D) Tracks showing the level of H3K9me3 on *rps19b* locus in control and *zfrps8* GKD showing reduction of heterochromatin. Source data are available online for this figure.

GKD overlapped with those in *SETDB1*, *Nup154*, and *e(z)* GKD (Fig. 3A; Dataset EV2). These genes, which are typically repressed in differentiated stages, are associated with quiescent chromatin (Fig. 3A). This overlap suggests that translation regulators and chromatin modulators co-regulate the silencing of a shared set of germ cell genes.

Our previous findings revealed that certain germ cell genes are not directly regulated by heterochromatin but rather become misexpressed due to their localization within topologically associating domains (TADs) (Kotb et al, 2024). Genes positioned within these domains appear especially vulnerable to global shifts in chromatin modifications or changes in nuclear architecture (Jerković et al, 2020; Kotb et al, 2024). For instance, while the gene *blanks* is not directly targeted by the heterochromatin-depositing enzyme SETDB1, its silencing depends on intact genome architecture mediated by NPCs (Kotb et al, 2024; Sarkar et al, 2023). In contrast, *rpS19b* carries heterochromatic marks on its promoter and gene body (Kotb et al, 2024; Sarkar et al, 2023). Our previous work demonstrated that disrupting heterochromatin or NPCs, which are interlinked through a feedback loop, leads to compromised genome architecture and a failure to silence these germ cell genes (Sarkar et al, 2023; Kotb et al, 2024). To further explore this connection, we mapped the genes dysregulated upon loss of ribosome-level regulators in relation to TADs. This analysis revealed that these genes predominantly reside within TADs and are silenced in differentiated stages (Fig. 3B).

Given their co-regulation of target genes and their similar phenotypes, we hypothesized that *zfrp8*, *mio*, and *raptor* GKD could perturb chromatin state and genome organization, similar to GKD of *SETDB1*, *Nup154*, and *e(z)*. To test this, we performed Cleavage Under Targets and Release Using Nuclease (CUT&RUN) assays to assess levels of the H3K9me3 and H3K27me3 chromatin marks, deposited by SETDB1 and E(z), respectively, in control and *zfrp8* GKD ovaries (Skene and Henikoff, 2017). We analyzed H3K9me3 and H3K27me3 chromatin marks in WT and *zfrp8* GKD ovaries to define 10 genome-wide states based on their distribution patterns. Our analysis revealed a significant decrease in both H3K9me3 and H3K27me3 levels on a subset of genes in *zfrp8* GKD ovaries (Fig. 3C; Dataset EV3). However, we also observed a redistribution of these marks, with their loss at specific loci accompanied by their gain elsewhere in the genome (Fig. 3C). Notably, the loss of H3K9me3 was particularly evident at genes with promoter-associated H3K9me3 (Fig. EV3A,B). Among these, *rpS19b* exhibited a striking reduction in H3K9me3 levels, consistent with its transcriptional upregulation in *zfrp8* GKD ovaries (Fig. 3D).

These findings suggest that maintaining adequate ribosome-level regulators is essential for maintaining proper chromatin state.

## Ribosome biogenesis promotes translation of the nucleoporin and TORC1 regulator Nup44A

As we found that proper ribosome levels, TORC1 activity, and translation are crucial to silence germ cell genes, we sought to identify the targets that could be translationally affected when ribosome-level regulators are depleted. To investigate this, we performed a polysome-seq analysis on ovaries from *zfrp8* GKD and control flies (Fig. 4A) (McCarthy et al, 2022; Breznak et al, 2023). We observed a reduction in the levels of the small (40S) and large (60S) ribosomal subunits, and monosomes (80S), in *zfrp8* GKD compared to WT, consistent with its role in ribosome biogenesis (Fig. 4B) (Breznak et al, 2023). By comparing polysome-associated RNAs to total RNAs, we identified mRNAs whose translation was disrupted in *zfrp8*GKD ovaries relative to wild-type controls (Fig. 4C) (Ernlund et al, 2018). Our analysis revealed that 536 transcripts were translationally downregulated and 509 were translationally upregulated in *zfrp8* GKD ovaries, indicating widespread translational dysregulation.

Gene ontology (GO) analysis of the upregulated targets, included proteins involved in siRNA-mediated pericentric heterochromatin and proteins associated with chromatin organization, such as Suppressor of hairy wing (Su(Hw)) and M1BP, suggesting potential feedback regulation triggered by the loss of heterochromatin (Fig. 4C,D; Dataset EV3) (Bag et al, 2021; Baxley et al, 2011). In addition, genes involved in stem cell division were upregulated, indicating an impaired transition from a GSC-like program to an oocyte program.

Conversely, GO analysis of translationally downregulated transcripts revealed a significant reduction in ribosomal components (Fig. 4D), consistent with previous reports that loss of ribosome biogenesis components decreases ribosomal protein synthesis (Martin et al, 2022; Breznak et al, 2023). However, *rpS19b* was not among the downregulated ribosomal proteins, consistent with the observation that it is ectopically expressed upon loss of *zfrp8*. Additionally, visual inspection of the downregulated targets identified genes such as *Nup44A*, which encodes a nuclear pore protein and *lola-like/batman*, which belongs to a family of BTB/POZ domain and interacts with both Polycomb and Trithorax complexes (Dataset EV3) (Senger et al, 2011; Faucheux et al, 2003). We found that independent GKD of either *Nup44A* or *lola-like/batman* resulted in egg chambers that do not grow and sterility,

**A**

Polysome purification

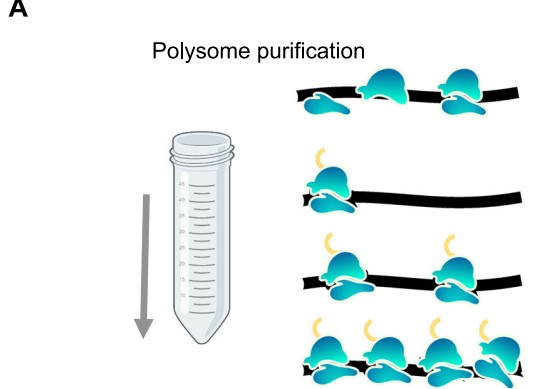

**B**

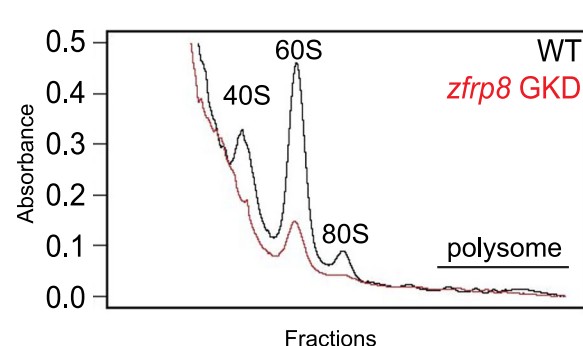

**C**

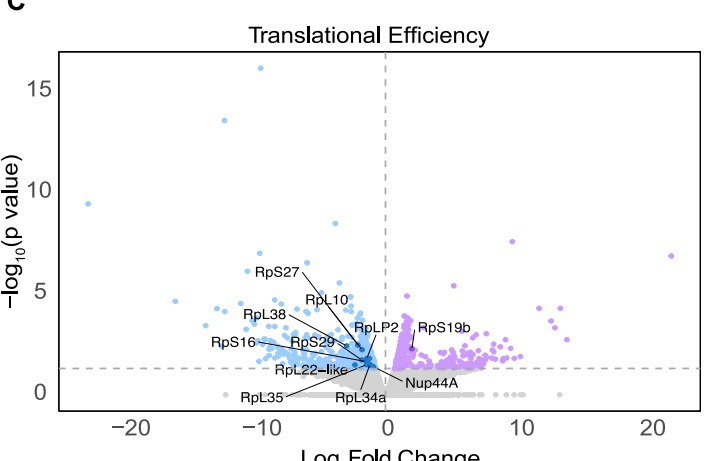

Translational Efficiency

**D**

| Gene Ontology of Translationally upregulated genes (p<0.05) | |
|---|---|
| **Process** | **Fold enrichment** |
| siRNA-mediated pericentric heterochromatin formation (GO:0140727) | 40 |
| stem cell division (GO:0017145) | 5 |
| regulation of mitotic cell cycle phase transition (GO:1901990) | 4 |
| **Gene Ontology of Translationally downregulated genes (p<0.05)** | |
| cytoplasmic translation (GO:0002181) | 8 |
| mitochondrial electron transport (GO:0006120) | 4 |

**E**

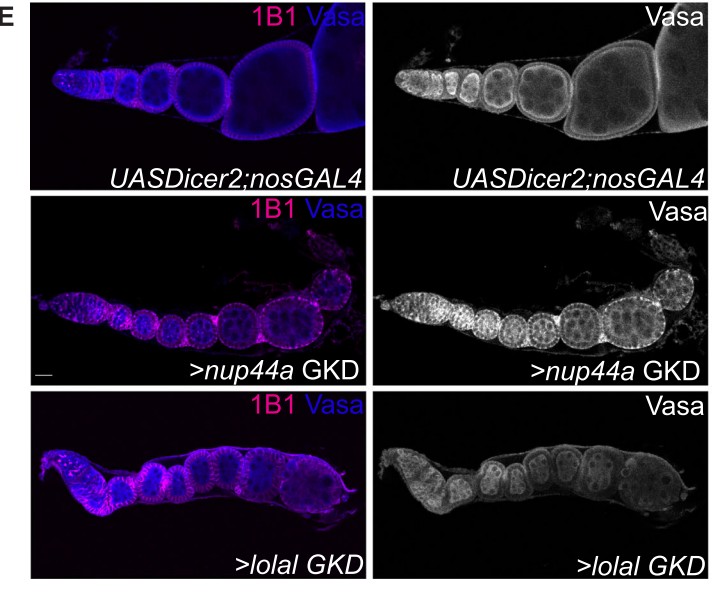

**F**

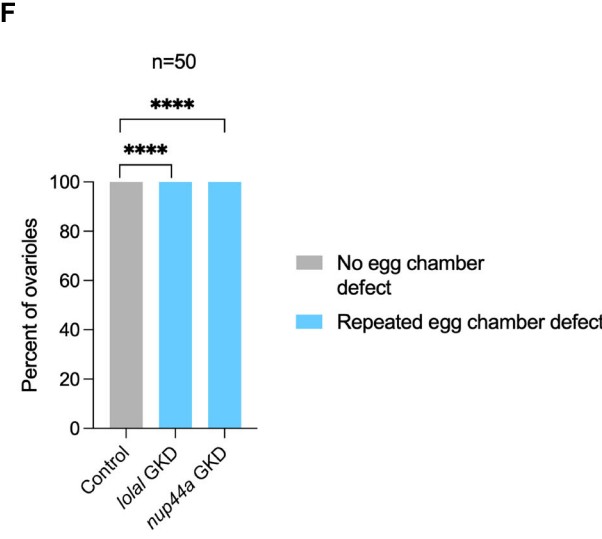

**Figure 4. Adequate ribosome levels are required for the translation of an NPC component.**

(A) A schematic of the polysome isolation from control *nosGAL4* and *>zfrp8* GKD ovaries using sucrose gradients. (B) Graph depicting 40S, 60S subunits, monosomes, and polysomes isolated from control *nosGAL4* and *>zfrp8* GKD ovaries. *zfrp8* is required in the germline for the proper formation of 40S, 60S, and monosomes. (C) Graph showing the translation efficiency comparing translational targets of *>zfrp8* GKD ovaries compared to *nosGAL4* (control) varies. Downregulated translational targets upon depletion of *zfrp8* are shown in blue and upregulated in lilac. ($n = 2$ biological replicates; 300 ovaries for zfrp8 GKD and 100 ovaries for WT). Differences in $\log_2$ fold change were assessed using a likelihood ratio test, and *P* values were corrected using the Benjamini–Hochberg false discovery rate (FDR). (D) Gene ontology of translationally upregulated and downregulated targets. (E) Ovarioles stained for Vasa (blue, shown in grayscale on right) and 1B1 (magenta). In *UAS-Dicer2;nosGAL4* controls, egg chambers grow. In contrast, GKD of *Nup44A* or *lolal* resulted in egg chambers that failed to grow and degenerated during oogenesis. (F) Quantitation of percent ovarioles with lack of egg chamber growth phenotype in *Nup44A* and *lolal* GKD, compared to control ovaries. Statistics: Fisher's exact test; $n = 50$ ovarioles per genotype; $P < 0.0001$ for *Nup44A* GKD and $P < 0.0001$ for *lolal* GKD. Source data are available online for this figure.

phenocopying loss of ribosome regulators and heterochromatin components (Fig. 4E,F). Thus, depletion of *zfrp8* results in translational dysregulation of components that regulate either proper chromatin state or genome architecture during oocyte specification.

## Nup44A is critical for silencing germ cell genes

Seh1, the mammalian homolog of Drosophila Nup44A, is a multifunctional protein essential for NPC assembly and TORC1 signaling regulation (von Appen et al, 2015; Senger et al, 2011). Within the NPC, Nup44A is a core component of the Nup107-160 subcomplex, a structural scaffold critical for NPC assembly, nuclear-cytoplasmic transport, and chromatin organization (Walther et al, 2003). This complex is particularly important during development, where it supports chromatin interactions and epigenetic regulation (Gozalo et al, 2020). Loss of Nup44A disrupts nuclear basket assembly (Senger et al, 2011). Beyond its structural role in the NPC, Nup44A also modulates TORC1 signaling, integrating nuclear transport with metabolic regulation (Senger et al, 2011). This dual function positions Nup44A as a crucial link between nuclear transport, metabolic signaling, and epigenetic regulation during development. Given its central role, we focused on Nup44A for further analysis.

In Drosophila, *Nup44A* mutants are homozygous viable but largely sterile due to a loss of oocyte specification (Senger et al, 2011). Since Zfrp8 regulates germ cell genes and is required for Nup44A translation, we hypothesized that Nup44A is similarly essential for silencing germ cell genes. To test this, we depleted *Nup44A* in the germline and assessed egg chamber growth and the expression of the *rpS19b::GFP* reporter. We found that GKD of *Nup44A* resulted in RpS19b::GFP expression in differentiated egg chambers and led to growth-arrested egg chambers, phenocopying the loss of heterochromatin components and ribosome regulators (Fig. 5A,B). To further investigate whether *Nup44A* regulates germ cell genes, we performed RNA-seq on *Nup44A* GKD ovaries, comparing gene expression profiles with those of ribosome regulators *zfrp8*, *mio*, *raptor*, and heterochromatin regulators *SETDB*, *Nup154*, and *e(z)*. We identified 2375 upregulated genes (Fig. 5C,D; Dataset EV4). Out of the 658 genes upregulated in *zfrp8*, *mio*, *raptor*, *SETDB1*, *Nup154*, and *e(z)* GKD, 97% are also upregulated in *Nup44A* GKD, including *rpS19b* (Fig. 5C,D; Dataset EV4). These findings establish Nup44A as a key regulator of germ cell gene silencing, linking NPC function to ribosome-mediated translational control during oocyte specification.

Previous studies had shown that NPC components are frequently enriched at TAD boundaries and that a large fraction of silenced germ cell genes reside within these domains (Kotb et al, 2024). We also found that genes dysregulated by ribosome-level regulators are predominantly located within TADs (Fig. 3B). Building on this, we asked whether genes affected by *Nup44A* depletion were also situated within TADs. By mapping genes dysregulated upon loss of *Nup44A*, we observed that these germ cell genes, much like those regulated by SETDB1 and the NPC component Nup154, are located within TADs (Fig. 5E). To investigate whether *Nup44A* regulates chromatin state, we performed CUT&RUN analysis for the heterochromatin marks H3K27me3 and H3K9me3 in *Nup44A* GKD ovaries. Similar to *zfrp8* depletion, we observed significant alterations in H3K27me3 and H3K9me3 levels, with their loss at specific loci accompanied by their gain elsewhere in the genome (Figs. 5F and EV4A,B). Notably, the loss of H3K9me3 was particularly evident at genes with promoter-associated H3K9me3, such as *rpS19b* (Fig. 5G). These findings reinforce the role of *Nup44A* in maintaining proper chromatin state and genome architecture, further supporting the model that ribosome biogenesis and NPC integrity work together to establish the epigenetic landscape necessary for oocyte specification.

## Nup44A supports a feedforward loop between TORC1 signaling and ribosome biogenesis to drive oogenesis

We next sought to determine how *Nup44A* translation is regulated by ribosome levels. We previously observed that terminal oligopyrimidine (TOP)-containing RNAs are translationally downregulated upon reduction of ribosome biogenesis in the GSCs (Martin et al, 2022). These motifs are known to coordinate the translation of specific proteins in response to TORC1 activity (Thoreen et al, 2012; Lahr et al, 2017; Fonseca et al, 2018; Philippe et al, 2018). Indeed, upon GKD of *zfrp8*, we observed that several ribosomal proteins are translationally downregulated (Fig. 4C,D). We hypothesized that the downregulation of *Nup44A* mRNA translation could be mediated by a potential TOP motif within its 5′ untranslated region (5′ UTR). To test this, we first asked if *Nup44A* 5′ UTR contained a TOP motif. Cap analysis of gene expression sequencing (CAGE-seq) data from total mRNA from the ovary was used to determine transcription start sites (TSSs) to accurately determine the 5' end of transcripts. TOP sequences start with a C or U base and contain a run of pyrimidine bases (Chen et al, 2014). We found that *Nup44A-RA* transcript was transcribed during oogenesis and the 5'UTR initiated with a C base and contained a stretch of pyrimidines (Fig. 6A). Thus, *Nup44A* contains all the hallmarks of a TOP motif-regulated transcript.

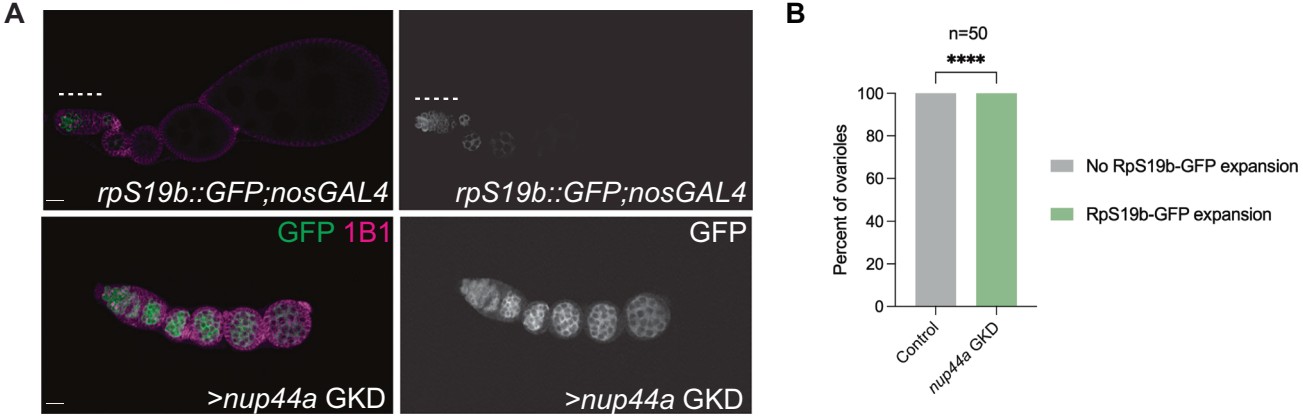

**A**

rpS19b::GFP;nosGAL4

rpS19b::GFP;nosGAL4

GFP 1B1

GFP

>nup44a GKD

>nup44a GKD

**B**

n=50

****

Percent of ovarioles

Control

nup44a GKD

No RpS19b-GFP expansion

RpS19b-GFP expansion

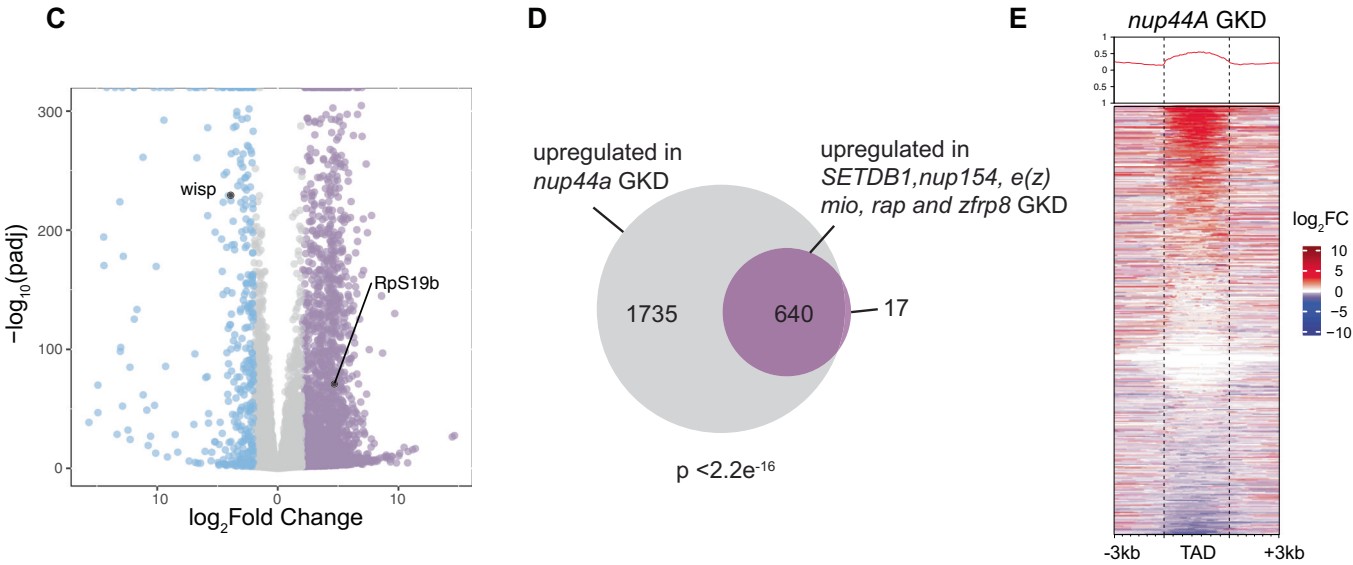

**C**

wisp

RpS19b

−log$_{10}$(padj)

log$_2$Fold Change

**D**

upregulated in
nup44a GKD

upregulated in
SETDB1,nup154, e(z)
mio, rap and zfrp8 GKD

1735

640

p <2.2e$^{-16}$

**E**

nup44A GKD

log$_2$FC

10
5
0
−5
−10

-3kb   TAD   +3kb

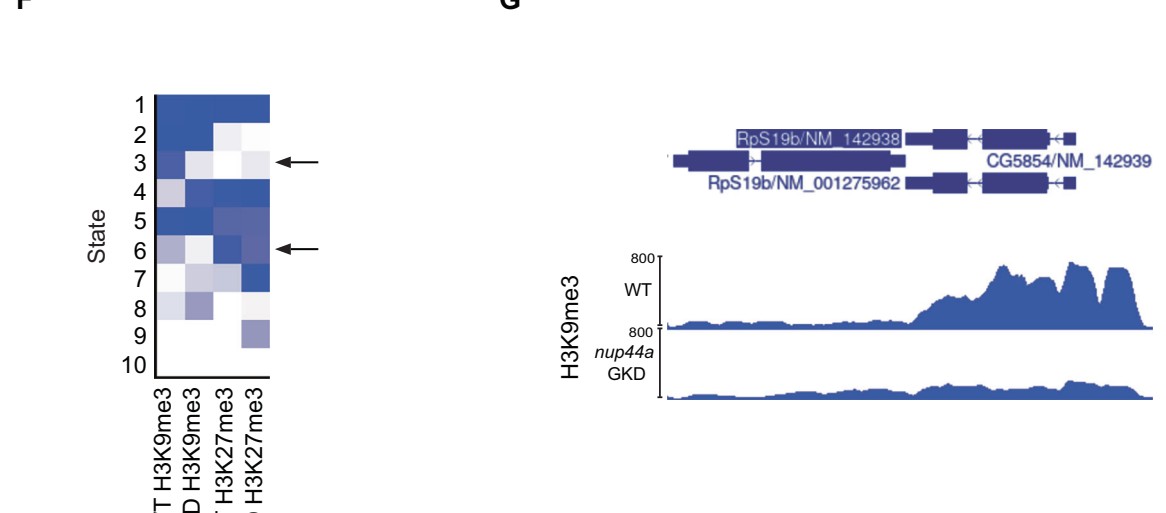

**F**

State

1
2
3
4
5
6
7
8
9
10

WT H3K9me3
zfrp8 GKD H3K9me3
WT H3K27me3
zfrp8 GKD H3K27me3

**G**

RpS19b/NM_142938

CG5854/NM_142939

RpS19b/NM_001275962

H3K9me3

800

WT

800

nup44a
GKD

**Figure 5.   Nup44A is required for silencing germ cell genes and promoting a stable epigenetic state.**

(A) Ovarioles stained for GFP (green, shown in grayscale on right) and 1B1 (magenta). In *rpS19b::GFP;nosGAL4* controls, GFP is expressed in undifferentiated germ cells and early cysts and then silenced. In contrast, GKD of *Nup44A* resulted in egg chambers that ectopically expressed RpS19b::GFP (white dashed line), failed to grow, and degenerated during oogenesis. (B) Quantitation of percent ovarioles with RpS19b::GFP expression. Statistics: Fisher's Exact Test; $n = 50$ ovarioles per genotype; $P < 0.0001$ for *Nup44A* GKD. (C) Volcano plots of $-\log_{10}$ adjusted $P$ value vs. $\log_2$ fold change (FC) of *nosGAL4> nup44a* GKD ovaries vs *nosGAL4* (control) showing significantly downregulated (blue) and upregulated (lilac) genes in *nup44a* GKD ovaries (FDR < 0.05 and genes with twofold or higher change were considered significant). Differential expression was calculated using DESeq2 with two biological replicates per genotype (Benjamini–Hochberg correction). (D) Venn diagram of upregulated genes from RNA-seq of >Nup44A compared to ribosome-level and chromatin regulators in Fig. 3. Statistics: Hypergeometric test $P < 2.2e^{-16}$. (E) RPKM-normalized tracks showing the level of H3K9me3 on *rps19b* locus in control and *Nup44A* GKD showing reduction of heterochromatin. (F) A chromatin model for H3K9me3 and H3K27me3 regulation in WT and >*Nup44A* GKD ovaries. In *Nup44A* GKD, H3K9me3 levels are reduced on promoters in state 3. (G) RPKM-normalized tracks showing the level of H3K9me3 on *rps19b* locus in control and *Nup44A* GKD showing reduction of heterochromatin. Source data are available online for this figure.

To investigate whether the TOP motif contributes to the regulation of translation in vivo during the cyst stages, we crossed a previously characterized TOP reporter line with *zfrp8* and *mio* GKD and monitored its translation levels in the cyst stages. Under normal conditions, the TOP reporter was strongly expressed in the cysts, reflecting its dependence on TORC1 activity (Fig. 6B). However, upon GKD of *zfrp8* or *mio*, GFP expression from the TOP reporter was significantly reduced, suggesting that both *zfrp8* and *mio* are required for the efficient translation of TOP motif-containing RNAs (Fig. 6B,C). As *Nup44A* is also implicated in the TORC1 pathway (Senger et al, 2011), we asked if GKD of *Nup44A* would also result in loss of TOP translation. Similar to the results with *zfrp8* and *mio* GKD, we found that *Nup44A* GKD led to a marked downregulation of GFP expression from the TOP reporter, further supporting the idea that ribosome-level regulators mediate the translation of TOP motif-containing RNAs (Fig. 6B,C). Overall, our data reveal that translation of the TOP-containing *Nup44A* mRNA is reduced when ribosome biogenesis is impaired.

## Discussion

### Translation regulates epigenetic reprogramming during oocyte specification

During oogenesis, germ cells undergo a tightly regulated differentiation process to give rise to a specified oocyte, which will mature into an egg capable of launching the next generation (Spradling et al, 2011; Lehmann, 2012). This process, termed germ cell-to-maternal transition, requires the silencing of germ cell genes and the activation of maternal genes, ensuring a stable and irreversible cell fate commitment (Sarkar et al, 2023; Kotb et al, 2024; Blatt et al, 2021). Here, we demonstrate that translation, supported by ribosome biogenesis and TORC1 activity, plays a pivotal role in regulating this epigenetic reprogramming. We identified ribosome biogenesis factors (Zfrp8, Bystin), TORC1 components (Raptor, Mio), and a translation-related factor eEF1α1 as critical regulators of gene silencing during oogenesis. Our findings suggest that the timing of the germ cell-to-maternal transition is mediated by TORC1 activity, which increases during oocyte specification to promote translation of key factors required for chromatin remodeling and gene silencing. This regulatory mechanism ensures that germ cell genes are repressed precisely as maternal programs are activated, establishing an irreversible commitment to oocyte fate.

Mechanistically, our findings suggest that ribosome biogenesis and TORC1 activity regulate nuclear architecture by influencing the translation of key NPC components, such as Nup44A (Senger et al, 2011; Gozalo et al, 2020), as well as chromatin through the regulation of the transcription factor Lolal, which interacts with chromatin modifiers (Faucheux et al, 2003). Since NPC assembly depends on the coordinated expression of its components, perturbations in ribosome biogenesis or TORC1 activity disrupt proper NPC formation, leading to defects in heterochromatin maintenance (Fig. 6D). Notably, our CUT&RUN analysis revealed that while H3K9me3 and H3K27me3 marks were reduced at certain loci, they were redistributed elsewhere in the genome. This suggests that most gene expression misregulation arises not solely from heterochromatin loss but from broader disruptions in genome organization. These findings support a model in which transient translational regulation establishes stable epigenetic states by promoting proper genome organization, ensuring irreversible cell fate transitions. By uncovering this translation-epigenetic axis, we reveal a previously unrecognized mechanism by which short-term fluctuations in translation are epigenetically encoded to drive long-term developmental decisions.

Although mutants defective in ribosome biogenesis or TORC1 activity fail to fully activate the oocyte transcriptional program, we observe that these cells still undergo partial differentiation. For example, while they retain *rps19b* expression and fail to grow, they are properly encapsulated into egg chambers, and their nuclei exhibit morphologies consistent with progression through early stages of oogenesis. This suggests that certain aspects of oocyte differentiation, such as morphological maturation and egg chamber formation, can proceed independently of TORC1-regulated translation. However, the full execution of oocyte fate, including silencing of germ cell genes and activation of maternal transcripts, appears to depend on sufficient translational output. Similar uncoupling has been observed in *egalitarian* mutants, which form egg chambers without specifying an oocyte (Navarro et al, 2004). These findings point to a modular control logic, in which translation-independent processes initiate oocyte differentiation, but translation-dependent mechanisms, such as chromatin remodeling and gene silencing, are required to commit to and maintain oocyte identity.

### Ribosome biogenesis orchestrates sequential transitions during germ cell differentiation into an oocyte

Ribosome biogenesis is a fundamental determinant of stem cell fate, balancing self-renewal and differentiation by regulating protein synthesis (Zhang et al, 2014; Sanchez et al, 2016; Khajuria et al, 2018). In many stem cell systems, high ribosome biogenesis

**A**

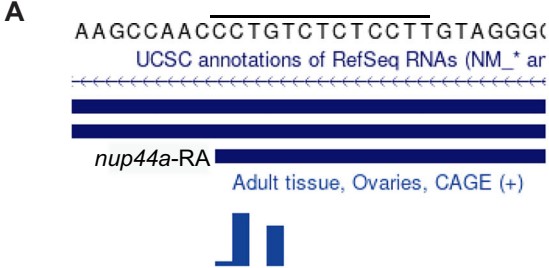

**B**

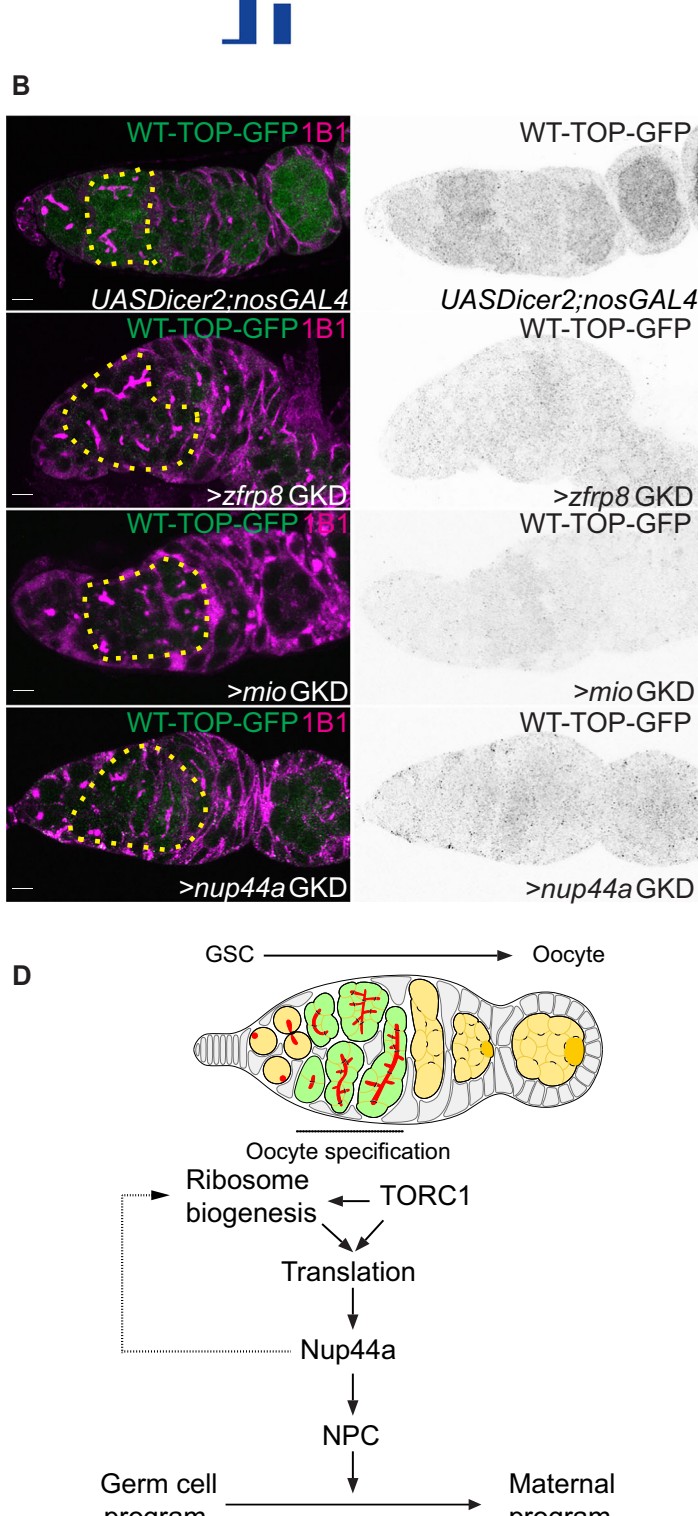

**C**

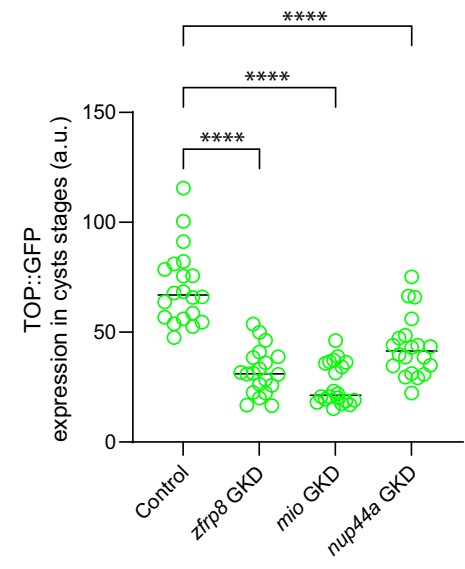

**D**

**Figure 6.** *Nup44A* contains a TOP-motif, and TOP-mRNA translation is sensitive to ribosome-level regulators in the cyst stages.

(A) CAGE-Seq analysis of the *Nup44A* transcript start site, showing an enrichment of pyrimidines, consistent with a terminal oligopyrimidine (TOP) motif, which is a feature of TORC1-sensitive mRNAs. (B) Germaria stained for GFP (green, shown in grayscale on right) and 1B1 (magenta). In *TOP-reporter::GFP* controls, reporter expression is observed in cyst stages. In contrast, GKD of *zfrp8*, *mio*, or *Nup44A* attenuated TOP reporter expression in the cyst stages compared to the control. (C) Quantification of GFP levels in the cyst stages of >*zfrp8* GKD, >*mio* GKD, and >*Nup44A* GKD compared to control ovaries. Loss of these factors results in reduced TOP reporter expression. Statistics: two-tailed *t* test; *n* = 50 ovarioles per genotype; *P* < 0.0001 for *zfrp8* GKD; *P* < 0.0001 for *mio* GKD; *P* < 0.0001 for *Nup44A* GKD. (D) Model illustrating how TORC1-driven translation and ribosome biogenesis regulate heterochromatin establishment during oocyte specification. In Drosophila, Zfrp8 promotes the translation of Nup44A, an NPC component, which is required for proper chromatin organization. H3K9me3 marks promoters of the germ cell genes to maintain their repression, while H3K27me3 regulates developmental genes. Loss of Zfrp8, TORC1 activity, or NPC components disrupts translation, leading to chromatin instability and gene misregulation, ultimately affecting oocyte fate commitment. This translation–epigenetic axis ensures the irreversible transition from germ cell to oocyte. Scale bars: 7.5 μm. Source data are available online for this figure.

sustains proliferation and maintains the undifferentiated state by coupling translational control with cell cycle progression (Breznak et al, 2022). However, as stem cells commit to differentiation, ribosome biogenesis is often downregulated, leading to selective translation of lineage-specific factors that drive cell fate transitions (Khajuria et al, 2018; Sanchez et al, 2016). This shift ensures that differentiation-associated gene programs are properly executed while preventing premature or inappropriate cell fate decisions.

We previously found that ribosome biogenesis plays a crucial role in orchestrating the stepwise differentiation of GSCs into oocytes (Martin et al, 2022). In GSCs, high ribosome biogenesis supports cell cycle progression by regulating factors that suppress P53, enabling continued cell division (Martin et al, 2022). However, upon GSC division, ribosome biogenesis levels decrease, coinciding with an increase in translation that drives the expression of meiotic genes and promotes early differentiation (Breznak et al, 2023). As germ cells progress into meiosis, translation levels increase (Sanchez et al, 2016), where it becomes essential for the synthesis of key factors, including Nup44A and chromatin regulators such as Lolal. These findings highlight a dynamic interplay between ribosome biogenesis and translation, where transient changes in ribosome levels dictate sequential waves of translation that activate differentiation programs in a controlled, stepwise manner.

While ribosome levels and translation dictate which mRNAs are translated efficiently, additional layers of regulation fine-tune protein synthesis during differentiation. We propose that trans-acting factors such as LARP, Pumilio, and Bruno act as RNA-binding proteins that license specific mRNAs for translation at distinct developmental stages (Flora et al, 2018b; Blatt et al, 2020). Thus, a combination of ribosome biogenesis, TORC1 activity, and translational control mechanisms ensures the timely expression of critical differentiation factors, allowing for a regulated transition from GSC self-renewal to oocyte specification.

### TORC1 integrates translational and epigenetic control to ensure oocyte fate commitment

TORC1 is a central regulator of cell growth and metabolism, integrating nutrient availability and cellular energy status to control translation (Laplante and Sabatini, 2012; Kim and Sabatini, 2004). In stem cell differentiation, TORC1 activity must be precisely modulated to balance self-renewal with lineage commitment (Iida and Lilly, 2004; Wei et al, 2014). In the germline, TORC1 activation promotes translation of factors required for cell cycle progression and early differentiation. However, our findings reveal that

TORC1's role extends beyond merely increasing protein synthesis; it also influences epigenetic reprogramming by regulating the translation of chromatin modifiers and nuclear pore components. Through this mechanism, TORC1 establishes the chromatin landscape necessary for differentiation, ensuring the timely repression of germline genes and activation of oocyte-specific programs.

Although we did not directly test canonical translational regulators such as eIF4E or 4EBP, our data argue that the key requirement is for general translation rather than selective regulation of specific transcripts. Supporting this, depletion of *rpS2*, a core ribosomal protein, phenocopies the loss of TORC1 (Cramton and Laski, 1994) (Jagut et al, 2013). These transcriptional defects mirror those seen upon loss of heterochromatin regulators such as SETDB1, *e(z)*, and *Nup154*, suggesting that insufficient protein synthesis broadly impairs the production of factors necessary for chromatin reorganization (Sarkar et al, 2023) (DeLuca et al, 2020). This link between translation and chromatin regulation suggests that adequate ribosome levels are required for proper heterochromatin establishment during oocyte specification. Supporting this, we observe significant changes in H3K27me3 and H3K9me3 marks upon depletion of *zfrp8*, reinforcing the idea that ribosome biogenesis is essential for chromatin remodeling during differentiation. However, as noted earlier, redistribution of chromatin marks suggests that the primary defect may lie in genome organization rather than a simple loss of heterochromatin (Iida and Lilly, 2004).

### Nup44A links NPC assembly and TORC1 activity to coordinate translational and epigenetic regulation

Nup44A, a component of the NPC, plays a crucial role in coordinating translational control and epigenetic reprogramming during oocyte specification (Senger et al, 2011). NPCs regulate nucleocytoplasmic transport and are essential for establishing chromatin organization, particularly through interactions with heterochromatin (Gozalo et al, 2020; Sarkar et al, 2023; Capelson et al, 2010). Our findings indicate that *Nup44A* translation is tightly regulated by ribosome biogenesis and TORC1 activity, positioning it as a key factor that synchronizes these two processes. TORC1-driven translation of *Nup44A* ensures proper NPC function, which in turn facilitates heterochromatin formation required for gene silencing. Loss of *Nup44A* disrupts nuclear architecture, leading to defects in the deposition of H3K9me3 and H3K27me3 marks, phenocopying the effects of TORC1 and ribosome biogenesis depletion.

While we focused on Nup44A in this study, our previous work indicates that other NPC components, such as Nup154 and Nup107, are also required for oocyte fate, suggesting that NPC integrity more broadly is essential for coordinating chromatin regulation during differentiation (Sarkar et al, 2023). Thus, rather than a unique role for Nup44A, we propose that proper NPC assembly acts as a functional platform that couples translational inputs to stable epigenetic outputs. By integrating ribosome-driven translation with nuclear organization, NPCs help synchronize transient metabolic cues with the long-term chromatin reprogramming required for the germ cell-to-maternal transition.

## Translation as a regulator of epigenetic memory

Our findings, together with previous work, highlight the conserved role of nutrient-sensing pathways in orchestrating chromatin architecture to regulate cell fate decisions. While fasting-induced chromatin reorganization in *C. elegans* is mediated through mTOR inhibition and RNA Pol I (Al-Refaie et al, 2024), we show that in Drosophila, TORC1 activation during oocyte specification drives a translation-dependent mechanism that establishes stable heterochromatin states. This suggests that metabolic inputs dynamically regulate chromatin architecture, either by remodeling genome organization in response to environmental cues, as seen in fasting (Al-Refaie et al, 2024), or by ensuring the irreversible silencing of developmental programs during differentiation, as observed in oogenesis. A key link between these processes may be the nucleolus and NPC, both of which are remodeled in response to translation and mTOR activity (Al-Refaie et al, 2024).

Overall, our work uncovers how transient translational changes impact long-term chromatin states, suggesting that ribosome biogenesis and translation could serve as central regulators of epigenetic memory across diverse biological contexts. This raises intriguing possibilities that metabolic shifts could influence chromatin states not just in development but also in disease and aging, opening new avenues for exploring how translation-mediated chromatin regulation shapes cellular identity and adaptation (Al-Refaie et al, 2024). Thus, by elucidating this intricate interplay, we provide a foundation for future studies on how translation-mediated chromatin regulation contributes to developmental robustness and cellular plasticity.

# Methods

### Reagents and tools table

| Reagent/resource | Reference or source | Identifier or catalog number |
| --- | --- | --- |
| **Experimental models** | | |
| UAS-Dcr2;nosGAL4 | Bloomington Drosophila Stock Center | 25751 |
| bamGAL4 | Bloomington Drosophila Stock Center | 80579 |
| nosGAL4;MKRS/TM6 | Bloomington Drosophila Stock Center | 4442 |

| Reagent/resource | Reference or source | Identifier or catalog number |
| --- | --- | --- |
| zfrp8 RNAi | Bloomington Drosophila Stock Center | 36581 |
| mio RNAi | Bloomington Drosophila Stock Center | 57745 |
| raptor RNAi | Bloomington Drosophila Stock Center | 34814 |
| bystin RNAi | Bloomington Drosophila Stock Center | 34876 |
| aramis RNAi | Bloomington Drosophila Stock Center | 32334 |
| RpS19b::GFP | McCarthy et al, 2022 | |
| **Recombinant DNA** | | |
| **Antibodies** | | |
| Mouse anti-1B1 | DSHB | AB_528070 |
| Rabbit anti-Vasa | Rangan Laboratory | |
| Chicken anti-Vasa | Rangan Laboratory | |
| Rabbit anti-GFP | Abcam | ab6556 |
| Rabbit anti-H3K9me3 | Active Motif | AB_2532132 |
| Mouse anti-H3K27me3 | Abcam | ab6002 |
| Rabbit anti-Egl | Lehmann Laboratory | |
| Mouse anti-NPC | BioLegend | AB_2565026 |
| Rabbit phospho-s6 | Rangan Laboratory | |
| Alexa 488 | Jackson Labs | 711-545-152 |
| Cy3 | Jackson Labs | 715-165-150 |
| Cy5 | Jackson Labs | 703-175-155 |
| **Oligonucleotides and other sequence-based reagents** | | |
| **Chemicals, enzymes, and other reagents** | | |
| **Software** | | |
| **Other** | | |
| Illumina NextSeq500 | Illumina | |
| Revvity NEXTFLEX Rapid Directional RNA-Seq Kit 2.0 | Revvity | NOVA-5198-02 |

## Drosophila stocks

This investigation utilized the following RNAi and mutant Drosophila lines: *zfrp8* RNAi (Bloomington #36581), *mio* RNAi (Bloomington #57745), *raptor* RNAi (Bloomington # 34814) *bystin* RNAi (Bloomington # 34876) and *aramis* RNAi (Bloomington #32334). The study also employed the RpS19b::GFP tagged line (McCarthy et al, 2022). Germline-specific drivers and double balancer lines used included: UAS-Dcr2;*nos*GAL4 (Bloomington #25751), *bam*GAL4 (Bloomington #80579), *mat*GAL4 (Bloomington #7062,7063), *nos*GAL4;MKRS/TM6 (Bloomington #4442).

## Drosophila husbandry

Fly crosses were cultivated at 25–29 °C, with dissections performed between 0 and 3 days post-eclosion. Fly food for stocks and crosses was prepared following the previously published laboratory protocol (summer/winter mix). Narrow vials (Fisherbrand Drosophila Vials; Fisher Scientific) were filled to ~10–12 mL (Flora et al, 2018a; Upadhyay et al, 2018).

## Dissection and immunostaining

Ovaries were extracted, and ovarioles were separated using mounting needles in PBS solution on ice. Samples were fixed for 12 min in 5% methanol-free formaldehyde. Ovaries underwent four 10-min washes in 0.5 mL PBT (1× PBS, 0.5% Triton X-100, 0.3% BSA) while nutating. Primary antibodies in PBT were applied and incubated overnight at 4 °C with nutation. Samples were then washed three times for 5–8 min each in 1 mL PBT. Secondary antibodies were added in PBT with 4% donkey serum and incubated at room temperature for 2–3 h. Samples underwent three 10-min washes in 1 mL of 1× PBST (0.2% Tween 20 in 1× PBS) and were incubated in Vectashield with DAPI (Vector Laboratories) for a minimum of 1 h before mounting.

## Antibodies

Primary antibodies utilized were: mouse anti-1B1 (1:20; DSHB), Rabbit anti-Vasa (1:5000; Rangan Laboratory), Chicken anti-Vasa (1:5000; Rangan Laboratory), Rabbit anti-GFP (1:2000; abcam, ab6556), Rabbit anti-H3K9me3 (1:500; Active Motif, AB_2532132), Mouse anti-H3K27me3 (1:500; abcam, ab6002), Rabbit anti-Egl (1:1000; Lehmann Laboratory), Rabbit anti-Blanks (1:2000; Sontheimer lab), Mouse anti-NPC (1:2000; BioLegend, AB_2565026). Secondary antibodies (Alexa 488 from Molecular Probes, Cy3 and Cy5 from Jackson Labs) were used at a 1:500 dilution.

## Fluorescence microscopy

Ovaries were mounted on slides and imaged using Zeiss LSM-710 and LSM-880 confocal microscopes under ×20, ×40, and ×63 oil objectives with a pinhole set to 1 airy unit. Image processing was conducted using Fiji, with gain adjustment and cropping performed in Photoshop.

## RNA extraction and TURBO DNase treatment

Ovaries were dissected into PBS, transferred to RNase-free microcentrifuge tubes, and flash-frozen in 100 μl Trizol at −80 °C. Samples were lysed using a plastic disposable pestle, with Trizol added to 1 mL total volume. After 5 min at room temperature, samples were centrifuged (20 min, >13,000× g, 4 °C). The supernatant was mixed with 500 μl chloroform, incubated for 5 min at room temperature, and centrifuged (10 min, max speed, 4 °C). The aqueous phase underwent ethanol precipitation with sodium acetate (10% of transferred volume) and 2–2.5 volumes of 100% ethanol at −20 °C overnight. RNA was pelleted by centrifugation (15 min, max speed, 4 °C), washed with 75% ethanol,

dried, and resuspended in 20–50 μl RNase-free water. Concentration was measured via nanodrop (Blatt et al, 2021).

## RNA-seq library preparation and analysis

Libraries were prepared using the Revvity NEXTFLEX Rapid Directional RNA-Seq Kit 2.0 (NOVA-5198-02). RNA was treated with Turbo DNase (TURBO DNAfree Kit, Life Technologies, AM1907) at 37 °C for 30 min. DNase was inactivated, and RNA was purified by centrifugation. Poly-A selection was performed on normalized RNA quantities. mRNA libraries were constructed, with quantity assessed via Qubit and quality via Bioanalyzer or Fragment Analyzer. Sequencing generated single-end 100 base pair reads on an Illumina NextSeq500.

Raw sequencing data quality was assessed using FastQC (v0.11.8), and adapter trimming was performed with Trim Galore! (v0.6.6). Reads were aligned to the *Drosophila melanogaster* dm6 reference genome (BDGP6.32, Ensembl 107) using the STAR aligner (v2.7.5b). For transcript quantification, Salmon (v1.2.1) was used in alignment-based mode. Genes with fewer than five total reads across all samples were excluded from downstream analysis.

Differential expression analysis was performed using the DESeq2 R package (v1.30.1). Genes were considered differentially expressed if they met an adjusted *P* value threshold of <0.05 and an absolute $\log_2$ fold change ≥2. Genes with less than 5 total reads across all samples were filtered out. All visualizations, including volcano plots and summary figures, were generated using the ggplot2 (v3.5.1) package in R (v4.2.0).

## DNA FISH with immunofluorescence

Ovaries were fixed, washed, and incubated with primary antibodies overnight. After secondary antibody incubation, samples underwent a series of washes with increasing formamide concentrations. Pre-denaturation was performed at varying temperatures, followed by probe hybridization overnight at 37 °C. Samples were washed, mounted in Vectashield, and imaged (Sarkar et al, 2023).

## CUT & RUN assay

Ovaries were permeabilized, incubated with antibodies overnight, and treated with pAG-MNase. DNA cleavage was induced with CaCl2, followed by RNase treatment and Proteinase K digestion. DNA was purified using magnetic beads and quantified via Qubit assay and Fragment analyzer.

Raw sequencing data were quality-checked using FastQC (v0.11.8) and trimmed for adapter sequences and low-quality bases using Trim Galore! (v0.6.6). Reads were aligned to the *Drosophila melanogaster* dm6 reference genome (BDGP6.32, Ensembl 107) using Bowtie2 (v2.2.8) and processed with samtools (v1.11) and Picard (v2.2.4) for sorting, indexing, quality filtering, and duplicate removal.

Chromatin state segmentation was performed using ChromHMM (v1.24). Binarized signal tracks were generated from aligned CUT&RUN data, and genome segmentation was executed using a multivariate Hidden Markov Model. A custom 10-state model was learned based on the input datasets, and the resulting chromatin state model was used to mark repressive histone modification gain and loss across the genome.

Signal tracks and enrichment heatmaps were visualized using deepTools (v2.1.0) and EnrichedHeatmap (v1.34.0). Additional processing and analysis steps incorporated bedtools (v2.29.2) and Subread (v2.0.1).

## Polysome seq

Polysome profiling of ovaries was carried out with modifications based on established protocols (Breznak et al, 2023). The frozen tissue was then homogenized in lysis buffer using a motorized pestle, with 20% of the lysate reserved for mRNA extraction and library preparation as previously described. The remaining lysate was loaded onto sucrose gradients (10–45%) supplemented with cycloheximide, using Beckman Coulter 9 of $16 \times 3.5$ PA tubes (#331372), and centrifuged at $35,000 \times g$ in an SW41 rotor for 3 h at 4 °C. After centrifugation, the gradients were fractionated using a density gradient fractionation system. RNA was extracted from the fractions with acid phenol–chloroform, precipitated overnight, and the resulting pellet was resuspended in 20 μl of water, treated with TURBO DNase, and used for library preparation as outlined above (McCarthy et al, 2022).

To assess differential translational efficiency, we analyzed RNA-seq data generated from polysome-associated and total mRNA (input) fractions from wild-type (WT) and knockout (KO) *Drosophila* samples. Samples were processed in the same way as bulk RNA-seq data, utilizing FastQC (v0.11.8) and Trim Galore! (v0.6.6). Gene-level counts were quantified using Salmon (v1.2.1).

Our analytical strategy was modeled on the statistical approach implemented in the RIVET algorithm (Ernlund et al, *BMC Genomics*, 2018), which quantifies translational efficiency as the interaction between condition (e.g., WT vs KO) and RNA fraction (polysome vs input). Specifically, we applied a generalized linear model to model both transcriptional and translational expression data simultaneously and then used a contrast that captures differential translation independent of transcription.

First, we constructed a DGEList object using the edgeR package (v3.42.4) with raw gene counts and grouped samples by condition and RNA fraction. Normalization was performed using the trimmed mean of M-values (TMM). A design matrix was generated to model the four experimental conditions: *WT_Input*, *KO_Input*, *WT_Polysomes*, and *KO_Polysomes*. Dispersion was estimated using empirical Bayes methods, and a negative binomial generalized linear model was fit to the data.

We then defined a contrast to isolate changes in translational efficiency: (KO_Polysomes − KO_Input) − (WT_Polysomes − WT_Input). This formulation tests whether the genotype alters the translational output of a gene beyond changes attributable to transcription alone.

We performed a likelihood ratio test (LRT) for the translational efficiency contrast, and genes with $P$ value < 0.05 were considered significantly translationally dysregulated.

## Data availability

Raw and unprocessed RNA-seq and CUT & RUN data are available in the Gene Expression Omnibus (GEO) databank under accession number GSE292611. https://www.ncbi.nlm.nih.gov/geo/query/acc.cgi?acc=GSE292611 Token: enmlkucylhkxfmb Publicly available

genome browser tracks were obtained of CAGE-seq data (generated by Chen et al, 2014) and viewed through the UCSC Genome Browser. The original CAGE-seq data from the ovaries were obtained from SRA under the accession number SRR488282. TADs are obtained from GSE89112 (Eagen et al, 2015).

The source data of this paper are collected in the following database record: biostudies:S-SCDT-10_1038-S44318-026-00697-0.

## Peer review information

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

## Acknowledgements

We thank members of the Rangan laboratory and Dr. Vernon Monteiro for their comments on the manuscript. We thank the Bloomington *Drosophila* Stock Center, Vienna *Drosophila* Resource Center, Berkeley *Drosophila* Genome Project Gene Disruption Project, and FlyBase for reagents and resources. PR is funded by the NIH/NIGMS (R35GM152236). This work was supported in part by the Bioinformatics for Next Generation Sequencing (BiNGS) shared resource facility within the Tisch Cancer Institute at the Icahn School of Medicine at Mount Sinai, which is partially supported by NIH grant P30CA196521, and the Icahn School of Medicine at Mount Sinai (ISMMS) Skin Biology and Disease Resource-based Center NIAMS P30 (AR079200). This work was also supported in part through the computational resources and staff expertise provided by Scientific Computing at the Icahn School of Medicine at Mount Sinai and supported by the Clinical and Translational Science Awards (CTSA) grant UL1TR004419 from the National Center for Advancing Translational Sciences. Research reported in this paper was also supported by the Office of Research Infrastructure of the National Institutes of Health under award numbers S10OD026880 and S10OD030463.

## Author contributions

**Noor M Kotb**: Data curation; Formal analysis; Validation; Visualization; Writing—original draft; Writing—review and editing. **Gulay Ulukaya**: Data curation; Formal analysis; Visualization. **Anupriya Ramamoorthy**: Data curation; Formal analysis; Investigation. **Lina Seojin Park**: Data curation; Formal analysis; Investigation. **Julia Tang**: Data curation; Formal analysis; Investigation. **Dan Hasson**: Data curation; Formal analysis. **Prashanth Rangan**: Conceptualization; Resources; Supervision; Funding acquisition; Writing—original draft; Project administration; Writing—review and editing.

Source data underlying figure panels in this paper may have individual authorship assigned. Where available, figure panel/source data authorship is listed in the following database record: biostudies:S-SCDT-10_1038-S44318-026-00697-0.

## Disclosure and competing interests statement

The authors declare no competing interests.

# Expanded View Figures

**Figure EV1. Ribosome-level regulators are required in the cyst stages for silencing germ cell gene *blanks* during oogenesis.**

(**A**) Quantification of the size of the egg chambers of control and *zfrp8* GKD, *bystin* GKD, *mio* GKD, *raptor* GKD and *eEF1a1* GKD ovaries. Graph shows that *zfrp8* GKD, *bystin* GKD, *mio* GKD, and *raptor* GKD's egg chambers do not grow compared to control ovaries. *eEF1a1* GKD did not lead to the formation of a third egg chamber therefore we were unable to quantitate. Statistics: Two-tailed *t* test; $n = 5$ ovarioles per genotype; $P = 0.0015$ for *aramis* GKD; $P = 0.0011$ for *zfrp8* GKD; $P = 0.0233$ for *mio* GKD; $P = 0.0010$ for *bystin* GKD; $P = 0.0138$ for *raptor* GKD. (**B**) Arbitrary unit (A.U.) quantification of RpS19b::GFP expression in *zfrp8* GKD, *bystin* GKD, *mio*GKD, *raptor* GKD and *eEF1a1* GKD compared to control ovaries. RpS19b::GFP persists in egg chambers of *zfrp8*GKD, *bystin* GKD, *mio* GKD, *raptor* GKD and *eEF1a1* GKD compared to the egg chambers of control. $n = 5$ ovarioles per genotype. Statistics: Two-tailed *t* test; $n = 50$ ovarioles per genotype; $P < 0.0001$ for *aramis* GKD $P < 0.0001$ for *zfrp8* GKD; $P < 0.0001$ for *mio* GKD; $P < 0.0001$ for *bystin* GKD; $P < 0.0001$ for *raptor* GKD; $P < 0.0001$ for *eEF1a1* GKD. (**C**) Wild-type control germarium stained for p-S6 (green, right in grayscale) and 1B1 (magenta). p-S6 marking TOR activity is expressed in the cyst stages. (**D**) Ovarioles stained for Blanks (green, shown in grayscale on right) and 1B1 (magenta). In *bamGAL4* controls, Blanks is expressed in undifferentiated germ cells (yellow arrows) and attenuated in egg chambers. In contrast, GKD of *zfrp8*, *mio*, or *eEF1a1* using *bamGAL4* caused egg chambers to ectopically express Blanks (white dashed lines), fail to grow, and degenerate during oogenesis. (**E**) Quantitation of percent ovarioles with Blanks expansion in control, *zfrp8* GKD, *bystin* GKD, *mio* GKD, *raptor* GKD and *eEF1a1* GKD ovaries. Statistics: Fisher's exact test; $n = 50$ ovarioles per genotype; $P < 0.0001$ for *zfrp8* GKD; p < 0.0001 for *mio* GKD; $P < 0.0001$ for *eEF1a1* GKD. Scale bars: (C-C1) 7.5 μm; (D-G1) 15 μm. Source data are available online for this figure.

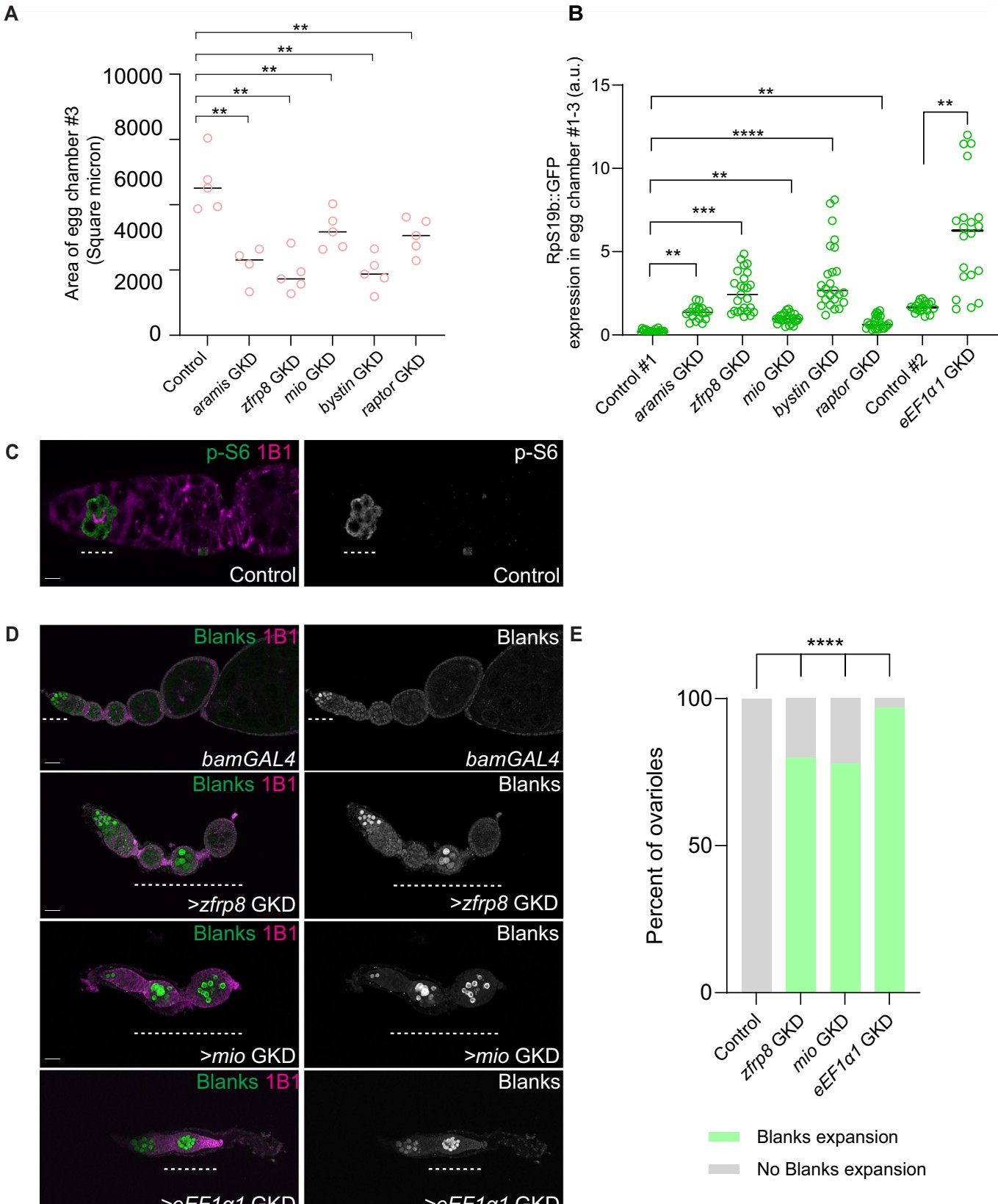

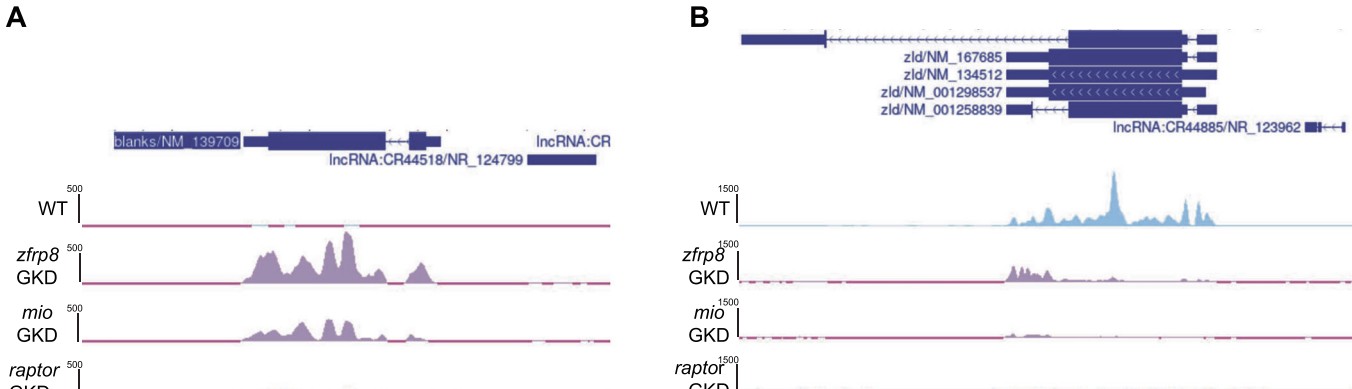

**Figure EV2.  Ribosome-level regulators are required for silencing germ cell genes during oogenesis.**

(A) RNA-seq tracks showing that *blanks* is upregulated upon >*zfrp8* GKD, >*mio* GKD and >*raptor* GKD (purple) compared to control *nosGAL4* (blue). (B) RPKM-normalized RNA-seq tracks showing that *zelda* is downregulated upon >*zfrp8* GKD, >*mio* GKD and >*raptor* GKD (purple) compared to control (*nosGAL4)* (blue).

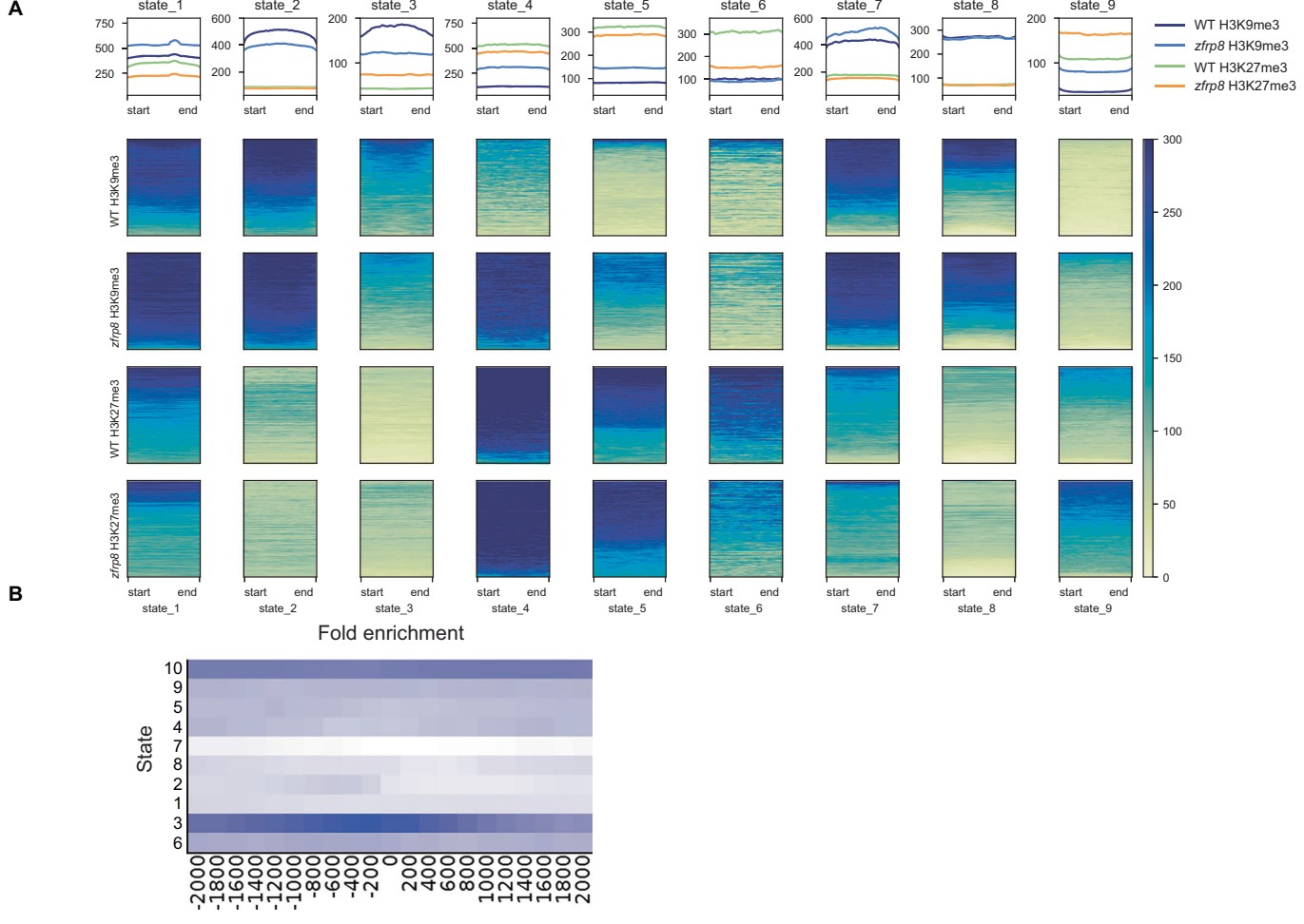

**Figure EV3. Ribosome-level regulators are required maintaining proper chromatin state.**

(A) A 10-state chromatin model depicting H3K9me3 and H3K27me3 distribution across the genome. This model illustrates H3K9me3 and H3K27me3 dynamics across 10 chromatin states, highlighting their distribution in different genomic regions. In *zfrp8* GKD, H3K9me3 is reduced on promoters in state 3. These findings reveal that ribosome biogenesis influences genome-wide epigenetic landscapes, coordinating transcriptional regulation during oogenesis. (B) In state 3, H3K9me3 is enriched around promoters. Source data are available online for this figure.

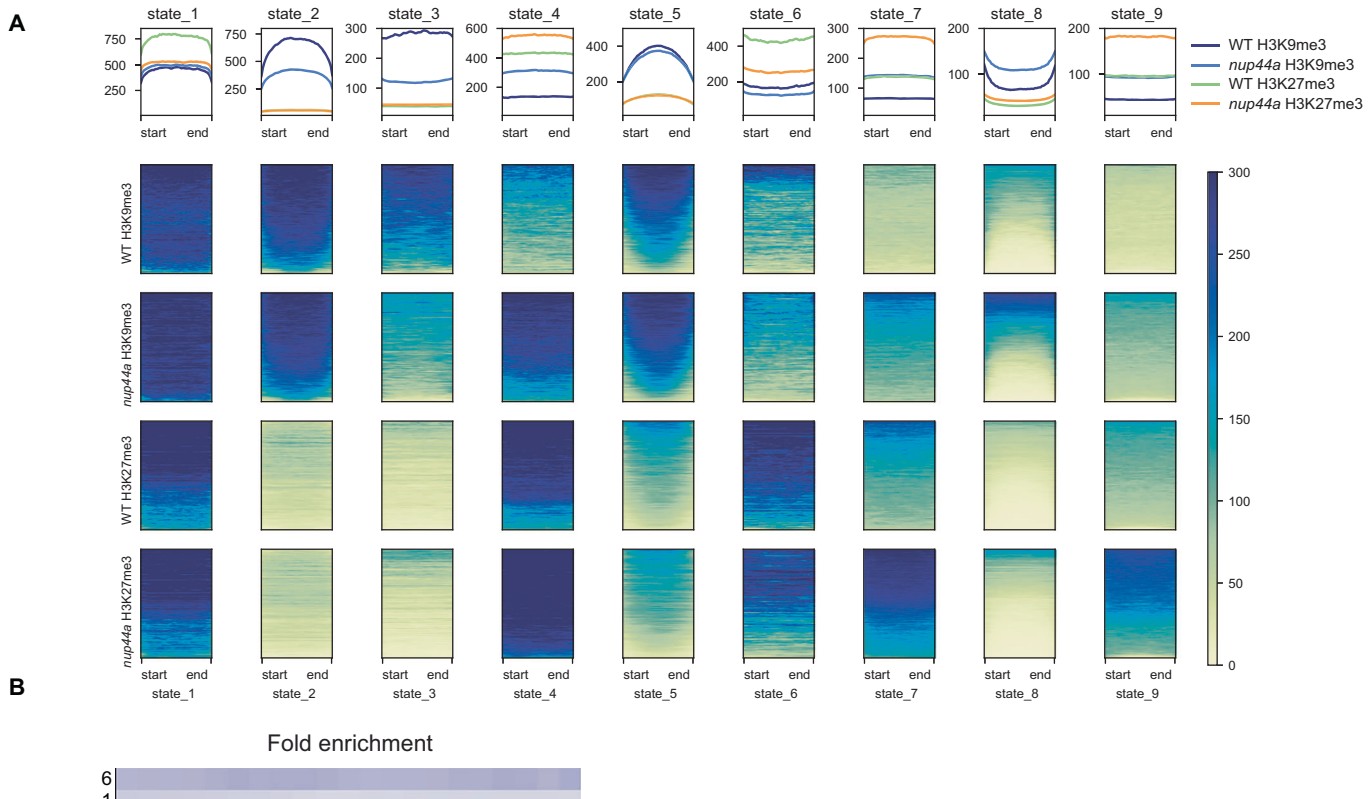

**Figure EV4.  Ribosome-level regulators are required for maintaining proper chromatin state.**

(**A**) A 10-state chromatin model depicting H3K9me3 and H3K27me3 distribution across the genome. This model illustrates H3K9me3 and H3K27me3 dynamics across 10 chromatin states, highlighting their distribution in different genomic regions. In *Nup44A* GKD, H3K9me3 is reduced on promoters in State 3 and H3K27me3 is redistributed across different states, indicating widespread chromatin reorganization. These findings reveal that Nup44A influences genome-wide epigenetic landscape. (**B**) In state 3, H3K9me3 is enriched around promoters, ensuring transcriptional repression of genes including *rps19b*. Source data are available online for this figure.

