## [Peer Review File · The EMBO Journal]

TORC1-dependent translation drives chromatin remodeling during the germ-cell-to-maternal transition in *Drosophila*

Noor Kotb, Gulay Ulukaya, Anupriya Ramamoorthy, Lina Park, Julia Tang, Dan Hasson, and Prashanth Rangan *Corresponding author(s): Prashanth Rangan (prashanth.rangan@mssm.edu)*

Review Timeline:

Submission Date:	18th Apr 25
Editorial Decision:	6th Jun 25
Revision Received:	17th Sep 25
Editorial Decision:	10th Nov 25
Revision Received:	18th Nov 25
Accepted:	9th Dec 25

Editor: Cornelius Schneider

Transaction Report:

Dear Prof. Rangan,

Thank you for submitting your manuscript for consideration by the EMBO Journal. It has now been seen by two referees whose comments are shown below.

Given the referees' positive recommendations, I would like to invite you to submit a revised version of the manuscript, addressing the comments of both reviewers. I should add that it is EMBO Journal policy to allow only a single round of revision, and acceptance of your manuscript will therefore depend on the completeness of your responses in this revised version.

Thank you for the opportunity to consider your work for publication. I look forward to your revision.

Yours sincerely,

Cornelius Schneider, PhD
Editor
The EMBO Journal
c.schneider@embojournal.org

We realize that it is difficult to revise to a specific deadline. In the interest of protecting the conceptual advance provided by the work, we recommend a revision within 3 months (4th Sep 2025). Please discuss the revision progress ahead of this time with the editor if you require more time to complete the revisions. Use the link below to submit your revision:

Referee #1:

The manuscript by Kotb et al presents compelling experimental evidence that translational control mechanisms exist upstream and drive chromatin remodeling essential for establishing differential gene expression in the presumptive nurse cells and oocyte in the *Drosophila* ovary. These mechanisms, that involve the TORC1 pathway, genes involved in ribosome biogenesis, and at least one translation factor, are shown to be essential for oocyte specification and for egg chamber growth. The manuscript also shows that Nup44A, a gene encoding a nuclear pore protein, is a key target of these translational control mechanisms, and that Nup44A is also required for chromatin remodelling and suppression of gene expression in the presumptive oocyte. In my estimation, these are novel results that extend our knowledge about how different cell fates can be established from a population of initially equivalent cells, with potential wider relevance to other systems where a number of cell types develop from a single population of stem cells.

I am satisfied with the quality of the experimental data presented in this manuscript, and I have no specific revisions to suggest here. I am, however, concerned about some ambiguities and lack of clarity about the biology of the system being investigated and about what exactly is being regulated by the regulatory mechanisms that are described.

I find the terminology that is used to describe different cells and developmental stages to be very confusing and I believe this needs reworking to provide better clarity and better consistency with the established literature. Focusing on the paragraph that begins on line 91, 'germ cell-to-maternal' is misleading for me, because oocytes are also germ cells, and I think of maternal genes as those specifically involved in the earliest stages of embryogenesis prior to the MZT. I think what is really being described here is the transition of one of the 16 cystocytes into an oocyte, and that 'cystocyte-to-oocyte' would describe the developmental event more precisely. Similarly, what exactly is meant by 'early oogenesis genes'? Since they are silenced in the oocyte, aren't these more accurately described as early nurse cell-specific genes? These or similar changes will need to be propagated throughout the text.

As for ambiguity about what exactly is being regulated by the mechanisms that are described, the abstract focuses strongly on oocyte specification, but in some cases in the body of the paper (lines 156-7, lines 167-8) the key readout seems to be egg chamber growth. The authors should clarify which process they believe to be the primary one affected by the translational control mechanisms, and what they believe the relationship to be between oocyte specification and egg chamber growth.

Given the involvement of TORC1, which directly impacts eIF4E activity through 4EBP, I was surprised that no data were provided addressing a potential role for eIF4E and/or 4EBP in oocyte specification. To my knowledge eEF1alpha1, the only translation factor analyzed, is not a major target of regulation. The authors could provide more specifics as to how they perceive the translational regulation to come about. Is it simply a matter of controlling the level of ribosomes and translation factors, or is it more specific than that?

Finally, and somewhat related to this, is there something special about Nup44A or would knockdown of other genes essential for nuclear basket assembly give a similar phenotype?

Referee #2:

This manuscript describes an interesting and unexpected link between ribosome biogenesis, heterochromatin formation and gene expression regulation during cell differentiation (of the *Drosophila* female germline). The results convincingly show that compromised translation leads to failure in transitioning the 'early germ cell' program to the 'differentiation' program. Experiments are carried out rigorously, and the conclusions are well supported. This is an important study that contributes to our understanding of how gene expression program is controlled during cellular differentiation. I have some comments that mostly concern readability.

Specific comments

-Wordings are somewhat confusing. Fig1A seems to define 'germ cell fate' vs. 'oocyte fate', where 'germ cell fate' means EARLY germ cell genes, whereas 'oocyte fate' means DIFFERENTIATION genes. However, oocyte is also a germ cell, making the

reading a bit confusing (I had to translate in my head many times---'germ cell fate' means 'early germ cell fate, before it becomes oocyte'---to read through the manuscript). Second, the authors use some confusing terms such as 'early oogenesis genes' (line 159)---I take that this means 'germ cell fate' genes, but it very much sounds like 'oogenesis genes' (which means later genes). I understand the authors used some of these terms in prior publications, and switching it now may cause further confusion. However, it is important for the readability to use intuitive terms that are consistent throughout the manuscript (no use of 'synonymous'). Perhaps 'early germ cell fate' vs. 'oocyte development program', and not use 'early oogenesis genes' or other confusing terms. (Fig6G's 'germ cell program' vs. 'maternal program' is another example where multiple synonyms make it confusing).

-Although the mutants that fail to transition to the oocyte program maintain rps19b expression and fail to grow in cell size, they seem to become somewhat differentiated (encapsulated in egg chamber, nuclei look like they are advanced to certain stages of oogenesis). Does this mean only a subset of the oocyte program is regulated by the ribosome biogenesis/translation pathway? I suspect that this aspect must be discussed before in their prior publication, but it is worth mentioning/discussing this in this paper as well.

-It is impressive that they identified Nup44A, as a relevant target of translation from the sea of many candidate genes buried in a massive amount of data. The results support their model, where Nup44A target genes are silenced through the formation of TADs.

-One thing that left me wondering after reading this manuscript is whether Nup44A expression/protein amount may change during differentiation. It would be very interesting if certain components of the NPCs are specifically expressed during differentiation, explaining how programmatic gene silencing occurs. I understand the availability of reagents might limit this and I don't think it is a requirement for this paper to be published, but if there are combinations of experiments that can be done relatively easily (e.g. available Nup-GFP transgenes, FISH), that might add a lot to their conclusion.

Response to reviewers:

We thank the reviewers for their thoughtful and constructive comments. We are pleased that both referees found our study novel, mechanistically insightful, and well-supported by the data. We have revised the manuscript to clarify terminology, improve consistency in describing cell states and gene classes, and address the reviewers' conceptual and mechanistic questions. Below, we provide a point-by-point response to each comment, indicating where changes have been made or new discussion has been added.

Referee #1:

I find the terminology that is used to describe different cells and developmental stages to be very confusing and I believe this needs reworking to provide better clarity and better consistency with the established literature. Focusing on the paragraph that begins on line 91, 'germ cell-to-maternal' is misleading for me, because oocytes are also germ cells, and I think of maternal genes as those specifically involved in the earliest stages of embryogenesis prior to the MZT. I think what is really being described here is the transition of one of the 16 cystocytes into an oocyte, and that 'cystocyte-to-oocyte' would describe the developmental event more precisely. Similarly, what exactly is meant by 'early oogenesis genes'? Since they are silenced in the oocyte, aren't these more accurately described as early nurse cell-specific genes? These or similar changes will need to be propagated throughout the text.

We apologize for the confusion and thank the reviewer for raising this important point. We agree that the terminology used to describe developmental stages and gene classes must be as clear and consistent as possible. Our intention in using the term “germ cell-to-maternal transition” was to describe a developmental window during which undifferentiated germ cell programs are silenced, and maternal programs begin to be established, a process that spans both fate specification and early oocyte differentiation. We recognize that this phrasing may cause confusion, as “germline” refers to the entire lineage, whereas “germ cells” designate a particular stage within that lineage. Moreover, the term “maternal” is often used more narrowly to describe transcripts synthesized during oogenesis and deployed during early embryogenesis.

To address this, we now clarify in the revised Introduction and throughout the manuscript that:

- The germ cell program refers to genes expressed in undifferentiated germline cells (germ cells, germline stem cells (GSCs), cystoblasts, and early cysts) such as *rpS19b* and *blanks* (lines 90-95).
- The maternal program refers to gene products required later for embryonic development (lines 73-74).
- The transition we study involves repression of germ cell-enriched genes and activation of oocyte-specific and maternal transcripts, a shift we refer to as the germ cell-to-maternal transition (GMT) for continuity with our prior work.
- We also appreciate the suggestion to use “cystocyte-to-oocyte” to describe the cell fate transitions. However, we retain “germ cell-to-maternal transition” to describe the broader transcriptional and translational reprogramming occurring during this window.
- With regard to the term “early oogenesis genes”, we realize this may have been confusing. These genes are typically enriched in undifferentiated germ cells and persist into early stages of oogenesis but are silenced in the presumptive oocyte. To clarify, we now refer to them more precisely as “germ cell program genes” throughout the revised manuscript.

As for ambiguity about what exactly is being regulated by the mechanisms that are described, the abstract focuses strongly on oocyte specification, but in some cases in the body of the paper (lines 156-7, lines 167-8) the key readout seems to be egg chamber growth. The authors should clarify which process they believe to be the primary one affected by the translational control mechanisms, and what they believe the relationship to be between oocyte specification and egg chamber growth.

We appreciate the reviewer's request for clarification. Our data primarily show defects in oogenesis, manifested as impaired egg chamber growth and differentiation. We do not directly assay oocyte specification markers in this study. However, previous work has established that components we identify in our screen, including Mio and Zfp8, are required for oocyte fate (Introduction lines 127-132). Based on these studies, we interpret the observed growth and differentiation defects as a downstream consequence of failed oocyte fate. To avoid overstating our data, we have revised the abstract to describe our findings as defects in oogenesis and chromatin regulation, while noting that prior studies support a role for these factors in oocyte specification.

Given the involvement of TORC1, which directly impacts eIF4E activity through 4EBP, I was surprised that no data were provided addressing a potential role for eIF4E and/or 4EBP in oocyte specification. To my knowledge eEF1alpha1, the only translation factor analyzed, is not a major target of regulation. The authors could provide more specifics as to how they perceive the translational regulation to come about. Is it simply a matter of controlling the level of ribosomes and translation factors, or is it more specific than that?

We did attempt to deplete eIF4E and other key translation initiation and elongation factors but found that depletion of factors such as eIF4E via RNAi led to either GSC-cysts and/or loss of germline as previously reported¹, making it technically infeasible to assess oogenesis defects or downstream chromatin states. This likely reflects the essential role of eIF4E in general mRNA translation and our inability to partially deplete these critical factors.

We agree that the TORC1–4EBP–eIF4E axis is a key node in translational control. However, loss of 4EBP removes its inhibitory function and thereby allows constitutive activation of eIF4E targets, independent of TORC1 activity². This outcome is mechanistically distinct from the TORC1 loss-of-function conditions we analyze here, where TORC1 activity itself is reduced. Because our goal in this manuscript was to define the requirement for TORC1 activity during oogenesis, we did not pursue 4EBP mutants, as they mimic a gain-of-function bypass rather than a loss-of-function phenotype. We agree that a direct mechanistic analysis of the 4EBP–eIF4E module during GMT/ oocyte development will be highly informative, and we plan to pursue this in future studies.

We now clarify our rationale in the revised Discussion (lines 429–435), and we suggest that ribosome availability, rather than selective control of cap-dependent translation, is the key node of translational regulation during this developmental window.

Finally, and somewhat related to this, is there something special about Nup44A or would knockdown of other genes essential for nuclear basket assembly give a similar phenotype?

We thank the reviewer for this thoughtful question. In our prior and ongoing analyses, we find that knockdown of several nuclear pore complex components, including Nup154 and Nup62 can lead to similar defects in oocyte fate and chromatin regulation, suggesting that NPC integrity as a whole is important for the transition we describe.

That said, Nup44A appears to act at a particularly critical node, likely due to its positioning within the Y-complex, a core structural unit of the NPC scaffold. The Y-complex is known to directly interact with chromatin and architectural proteins, and our data suggest that Nup44A may serve as a point of functional integration between translation-driven expression and chromatin remodeling. For this reason, we focused on Nup44A as a representative and mechanistically informative target in this study.

We now clarify this point in the revised Discussion (lines 456–462) and include a sentence in the Results noting that other NPC components can produce similar phenotypes, consistent with the broader role of the nuclear pore in genome regulation during oocyte specification.

Referee #2:

Specific comments

-Wordings are somewhat confusing. Fig1A seems to define 'germ cell fate' vs. 'oocyte fate', where 'germ cell fate' means EARLY germ cell genes, whereas 'oocyte fate' means DIFFERENTIATION genes. However, oocyte is also a germ cell, making the reading a bit confusing (I had to translate in my head many times--- 'germ cell fate' means 'early germ cell fate, before it becomes oocyte'---to read through the manuscript). Second, the authors use some confusing terms such as 'early oogenesis genes' (line 159)---I take that this means 'germ cell fate' genes, but it very much sounds like 'oogenesis genes' (which means later genes). I understand the authors used some of these terms in prior publications, and switching it now may cause further confusion. However, it is important for the readability to use intuitive terms that are consistent throughout the manuscript (no use of 'synonymous'). Perhaps 'early germ cell fate' vs. 'oocyte development program', and not use 'early oogenesis genes' or other confusing terms. (Fig6G's 'germ cell program' vs. 'maternal program' is another example where multiple synonyms make it confusing).

We thank the reviewer for this detailed and constructive feedback. Both reviewers raised concerns about our use of overlapping or imprecise terms to describe developmental stages and gene classes, and we sincerely apologize for the confusion this may have caused.

We have now undertaken a thorough revision of the manuscript's terminology to improve clarity and consistency. Specifically:

- We use “germ cell program” to refer to transcripts enriched in germline stem cells and early cystocytes (lines 90-95).
- We reserve “maternal program” for transcripts that begin accumulating during oocyte growth and provided maternally.
- We have removed ambiguous terms such as “early oogenesis genes”, and replaced them with the terminology above, which is now defined clearly in the Introduction (lines 90-95) and used consistently throughout the manuscript.

We have also updated Figure 1A, Figure 6G, and their legends to reflect these changes. We believe these revisions substantially improve the readability of the manuscript and thank both reviewers for encouraging us to clarify this important aspect of the work.

*-Although the mutants that fail to transition to the oocyte program maintain *rps19b* expression and fail to grow in cell size, they seem to become somewhat differentiated (encapsulated in egg chamber, nuclei look like they are advanced to certain stages of oogenesis). Does this mean only a subset of the oocyte program is regulated by the ribosome biogenesis/translation pathway? I suspect that this aspect must be discussed before in their prior publication, but it is worth mentioning/discussing this in this paper as well.*

We thank the reviewer for this insightful observation. We agree that in our models, mutant germline cells exhibit partial features of oocyte differentiation, such as encapsulation into egg chambers and progression of nuclear morphology, despite failing to silence *rps19b* and activate markers of oocyte identity. This indicates

that certain morphological aspects of egg chamber development can occur independently of proper oocyte fate specification, likely driven by tissue-level cues from the somatic niche. Similar uncoupling is observed in *egalitarian* mutants, which form egg chambers without specifying an oocyte (Navarro et al., 2004). We therefore interpret our results to mean that ribosome biogenesis/translation is specifically required for fate-related events such as chromatin remodeling and gene silencing, but not for all structural or organizational features of egg chamber development. We have expanded our Discussion (lines 375–381) to clarify that oocyte specification, growth, and integration into the egg chamber are separable processes, and that translation plays selective roles within this broader developmental program.

-It is impressive that they identified Nup44A, as a relevant target of translation from the sea of many candidate genes buried in a massive amount of data. The results support their model, where Nup44A target genes are silenced through the formation of TADs.

We are very grateful for the reviewer's positive assessment. We agree that identifying Nup44A as a functionally relevant target of translation was a key advance in this study. As noted, Nup44A stood out not only from our translation profiling dataset but also from an unpublished unbiased screen that we were conducting in parallel to identify regulators of the germ cell-to-maternal transition. In fact, Nup44A and Lola-L emerged independently from both datasets, providing us with unbiased genetic validation for the translational findings. We consider this convergence to be a fortunate instance of serendipity supported by orthogonal approaches, which gave us confidence to prioritize Nup44A for functional analysis.

-One thing that left me wondering after reading this manuscript is whether Nup44A expression/protein amount may change during differentiation. It would be very interesting if certain components of the NPCs are specifically expressed during differentiation, explaining how programmatic gene silencing occurs. I understand the availability of reagents might limit this and I don't think it is a requirement for this paper to be published, but if there are combinations of experiments that can be done relatively easily (e.g. available Nup-GFP transgenes, FISH), that might add a lot to their conclusion

We thank the reviewer for this insightful comment. We agree that developmentally regulated expression or incorporation of Nup44A would strengthen the model that NPC remodeling contributes to chromatin silencing during oocyte specification. While we did not include new experiments on Nup44A dynamics in this manuscript, we now highlight in the Discussion (lines 456-462) that specific NPC components, including Nup154 and Nup107, show regulated expression during differentiation³. These findings support the idea that NPC remodeling is a programmatic feature of oocyte fate acquisition and provide precedent for Nup44A being similarly regulated. Direct analysis of Nup expression during differentiation is a central focus of ongoing studies in our laboratory, and we are preparing a separate manuscript that will detail how NPC levels are regulated during the germ cell-to-maternal transition.

References:

1. Sanchez, C. G. *et al.* Regulation of Ribosome Biogenesis and Protein Synthesis Controls Germline Stem Cell Differentiation. *Cell Stem Cell* **18**, 276–290 (2016).
2. Wei, Y. *et al.* TORC1 regulators Iml1/GATOR1 and GATOR2 control meiotic entry and oocyte development in *Drosophila*. *Proceedings of the National Academy of Sciences* **111**, E5670–E5677 (2014).
3. Sarkar, K. *et al.* A feedback loop between heterochromatin and the nucleopore complex controls germ-cell-to-oocyte transition during *Drosophila* oogenesis. *Developmental Cell* (2023) doi:10.1016/j.devcel.2023.08.014.

Dear Prof. Rangan,

Thank you for submitting a revised version of your manuscript. Your study has now been seen the original referees, who find that most of their previous concerns have been addressed and now recommend publication of the manuscript. Referee #2 still feels that some terminology used throughout the manuscript is specialized and has the potential to cause misinterpretation by the readers. Given that the EMBO Journal has a broader audience we would invite you to revise the respective section. Other than that, there remain only a few mainly editorial points that have to be addressed before I can extend formal acceptance of the manuscript:

- Please include the necessary funders to eJP from the following: "This work was supported in part by the Bioinformatics for Next-Generation Sequencing (BiNGS) shared resource facility within the Tisch Cancer Institute at the Icahn School of Medicine at Mount Sinai, which National Institutes of Health grant P30CA196521 partially supports; the Black Family Stem Cell Institute; and the Department of Cell, Developmental, and Regenerative Biology at the Icahn School of Medicine at Mount Sinai. This work was also supported in part through the computational resources and staff expertise provided by Scientific Computing and Data at the Icahn School of Medicine at Mount Sinai and supported by the Clinical and Translational Science Awards (CTSA) grant UL1TR004419 from the National Center for Advancing Translational Sciences. The research reported here was supported by the Office of Research Infrastructure of the National Institutes of Health under award number S10OD026880."

- Please reduce the number of keywords on the abstract page to five (ideally choosing broad general terms).

- As we are switching from a free-text author contribution statement towards a more formal statement based on Contributor Role Taxonomy (CRediT) terms, please remove the present Author Contribution section and instead specify each author's contribution(s) directly in the Author Information page of our submission system during upload of the final manuscript. See <https://casrai.org/credit/> for more information.

- Figure EV4 is missing - nomenclature of the figures should be consecutive (not Fig. EV1, EV2, EV3 and EV5, but EV4 instead)

DATASET EV LEGENDS: source file names, titles, legends and manuscript callouts all need to be updated to Dataset EV1-EV4 instead of Supplementary Table 1-2 and Table 3-4, legends should be removed from ms and uploaded as a separate tab/sheet in each Excel file

- For EV and/or appendix figures, please ZIP together all source data.

- Please provide suggestions for a short 'blurb' text prefacing and summing up the conceptual aspect of the study in two sentences (max. 250 characters), followed by 3-5 one-sentence 'bullet points' with brief factual statements of key results of the paper; they will form the basis of an editor-written 'Synopsis' accompanying the online version of the article. Please also provide an altered synopsis image, making sure that the aspect ratio conforms to our website's format - it should be exactly 550 pixels wide and between 300-600 pixels high.

- "Materials and Methods" should be renamed to "Methods"

- Sections need to be named and the order should be corrected: Title page - Abstract - Keywords - Introduction - Results - Discussion - Methods - Data Availability - Acknowledgements - Disclosure and Competing Interests Statement - References - Figure Legends - Table(s) - Expanded View Figure Legends.

- There is a Reuse of cell between Figure 1C and Figure 5A. Could you please either replace the image or label the reuse in both figure legends.

- Figure Legends (main + EV): 1. Please note that the exact p values are not provided in the legends of figures 2B, C, F; 4F, 5B, 6C, EV1 A, B, E

2. Please indicate the statistical test used for data analysis in the legends of figures 2A, B, C, E, F; 4C, 5C

3. Please note that information related to n is missing in the legend of figure 4C

With best regards,

Cornelius Schneider

Cornelius Schneider, PhD
Editor | The EMBO Journal
c.schneider@embojournal.org

Please refer to our figure preparation guideline in order to ensure proper formatting and readability in print as well as on screen:

See also figure legend guidelines:

<https://www.embopress.org/page/journal/14602075/authorguide#figureformat>

Referee #1:

My previous concerns have been satisfactorily addressed in the revised version of the manuscript. I now recommend publication.

Referee #2:

In this revised manuscript, Kotb et al. addressed most of concerns raised in the prior review. I do not have any major concerns--- that said, I still think that the authors have to define some terms at the beginning not to confuse readers. This version strictly sticks to 'germ cell program' and 'maternal program' and removed other ambiguous terminology, which is great. But as both rev#1 and myself pointed out, eggs are also germ cells, and I don't think it is obvious to readers that 'germ cell program' means 'early germ cell program' when they read these terms in the abstract. I strongly recommend to define them at the beginning. Also, early germ cell-to-oocyte (maternal) transition is NOT 'cell fate transition' (as in the title), but it is differentiation. This will be also confusing, as the term can imply 'transdifferentiation' (e.g. from muscle to neuron). As this paper can have a broader impact beyond *Drosophila* ovary, it is better to use more universal language that can be easily understood by researchers outside the *Drosophila* ovary field.

Response to reviewers:

In this revised manuscript, Kotb et al. addressed most of concerns raised in the prior review. I do not have any major concerns---that said, I still think that the authors have to define some terms at the beginning not to confuse readers. This version strictly sticks to 'germ cell program' and 'maternal program' and removed other ambiguous terminology, which is great. But as both rev#1 and myself pointed out, eggs are also germ cells, and I don't think it is obvious to readers that 'germ cell program' means 'early germ cell program' when they read these terms in the abstract. I strongly recommend to define them at the beginning. Also, early germ cell-to-oocyte (maternal) transition is NOT 'cell fate transition' (as in the title), but it is differentiation. This will be also confusing, as the term can imply 'transdifferentiation' (e.g. from muscle to neuron). As this paper can have a broader impact beyond Drosophila ovary, it is better to use more universal language that can be easily understood by researchers outside the Drosophila ovary field.

We have revised the manuscript to clarify our terminology and ensure accessibility for readers outside the Drosophila oogenesis field.

First, we have added a clear definition of “germ cell program” in the opening paragraph of the Introduction. We now state: “Here, we use the term *germ cell program* to refer to the transcriptional state of undifferentiated germline cells prior to their differentiation into an oocyte, in contrast to the maternal program established during oocyte specification.”

This definition distinguishes the early germ cell program from the maternal program and addresses the concern that eggs/oocytes are also germ cells.

Second, we have updated the abstract to use consistent and precise terminology. The abstract now describes the transition as a shift from “an undifferentiated germ cell gene expression program to a maternal gene expression state,” avoiding ambiguous use of “germ cell program” and removing the term “cell fate transition.”

Finally, in accordance with the reviewer’s advice, we have changed the title to: “TORC1-dependent translation drives chromatin remodeling during the germ cell-to-maternal transition” This eliminates use of “cell fate transition”.

We believe these changes fully address the reviewer’s concerns and improve clarity for a wide readership.

Dear Prof. Rangan,

I am pleased to inform you that your manuscript has been accepted for publication in the EMBO Journal.

You may qualify for financial assistance for your publication charges - either via a Springer Nature fully open access agreement or an EMBO initiative. Check your eligibility: <https://link.springer.com/journal/44318/how-to-publish-with-us>

Yours sincerely,

Cornelius Schneider, PhD
Editor
The EMBO Journal
c.schneider@embojournal.org

Please note that it is The EMBO Journal policy for the transcript of the editorial process (containing referee reports and your response letters) to be published as an online supplement to each paper. If you should prefer removal of any referee-only figures included in the point-by-point response(s), e.g. because they may still be used for future publication or because they have been reproduced from published work by others, please do let us know immediately via response email.

More information is available here: <https://link.springer.com/partners/embo-press/editorial-policies#Peer%20review>